# Generation and Accumulation of Various Advanced Glycation End-Products in Cardiomyocytes May Induce Cardiovascular Disease

**DOI:** 10.3390/ijms25137319

**Published:** 2024-07-03

**Authors:** Takanobu Takata, Shinya Inoue, Togen Masauji, Katsuhito Miyazawa, Yoshiharu Motoo

**Affiliations:** 1Division of Molecular and Genetic Biology, Department of Life Science, Medical Research Institute, Kanazawa Medical University, Uchinada, Ishikawa 920-0293, Japan; 2Department of Pharmacy, Kanazawa Medical University Hospital, Uchinada, Ishikawa 920-0293, Japan; masauji@kanazawa-med.ac.jp; 3Department of Urology, Kanazawa Medical University, Uchinada, Ishikawa 920-0293, Japan; s-inoue@kanazawa-med.ac.jp (S.I.); miyazawa@kanazawa-med.ac.jp (K.M.); 4Department of Internal Medicine, Fukui Saiseikai Hospital, Wadanaka, Fukui 918-8503, Japan

**Keywords:** cardiovascular disease, ryanodine receptor 2 (RyR2), F-actin–tropomyosin filament, advanced glycation end-products (AGEs), mass, AGE pattern, traditional medicines, natural products, carbonyl trap, glyoxalase-1 (GLO-1)

## Abstract

Cardiomyocyte dysfunction and cardiovascular diseases (CVDs) can be classified as ischemic or non-ischemic. We consider the induction of cardiac tissue dysfunction by intracellular advanced glycation end-products (AGEs) in cardiomyocytes as a novel type of non-ischemic CVD. Various types of AGEs can be generated from saccharides (glucose and fructose) and their intermediate/non-enzymatic reaction byproducts. Recently, certain types of AGEs (*N*^ε^-carboxymethyl-lycine [CML], 2-ammnonio-6-[4-(hydroxymetyl)-3-oxidopyridinium-1-yl]-hexanoate-lysine [4-hydroxymethyl-OP-lysine, hydroxymethyl-OP-lysine], and *N*^δ^-(5-hydro-5-methyl-4-imidazolone-2-yl)-ornithine [MG-H1]) were identified and quantified in the ryanodine receptor 2 (RyR2) and F-actin–tropomyosin filament in the cardiomyocytes of mice or patients with diabetes and/or heart failure. Under these conditions, the excessive leakage of Ca^2+^ from glycated RyR2 and reduced contractile force from glycated F-actin–tropomyosin filaments induce cardiomyocyte dysfunction. CVDs are included in lifestyle-related diseases (LSRDs), which ancient people recognized and prevented using traditional medicines (e.g., Kampo medicines). Various natural compounds, such as quercetin, curcumin, and epigallocatechin-3-gallate, in these drugs can inhibit the generation of intracellular AGEs through mechanisms such as the carbonyl trap effect and glyoxalase 1 activation, potentially preventing CVDs caused by intracellular AGEs, such as CML, hydroxymethyl-OP, and MG-H1. These investigations showed that bioactive herbal extracts obtained from traditional medicine treatments may contain compounds that prevent CVDs.

## 1. Introduction

Cardiovascular disease (CVD) is a serious issue worldwide, including in China [1]. In China, about 4 million deaths are caused annually by CVDs, accounting for 40% of the overall deaths in the population [1]. Cardiomyocytes account for approximately 25–30% of the cardiac tissue and control beating [2]. The circulation of Ca^2+^ in the cardiomyocytes is regulated by the sarcoplasmic reticulum (SR), which contains sarco/endoplasmic reticulum Ca^2+^ ATPase 2α (SERCA2α) and ryanodine receptor type 2 (RyR2) [3,4,5,6,7]. Additionally, the combination of Ca^2+^ and troponin induces the movement of cardiac actin/myosin filaments, resulting in a heartbeat [8,9,10]. Cardiomyocyte dysfunction induces various cardiovascular diseases (CVDs), and its mechanisms can be divided into ischemic [11,12,13,14,15] or nonischemic types [11]. Because cardiomyocytes require a supply of oxygen, glucose, and Ca^2+^ from the blood, ischemia, which leads to atherosclerosis [12,13], Mönckeberg’s medial calcific sclerosis (MMCS) [14,15], and arteriocapillary sclerosis [16] can cause various forms of CVD. In contrast, non-ischemic CVDs, such as myocarditis [17,18], diabetic cardiomyopathy [19,20], cardiac tumors [21,22], and injury by anticancer agents against cardiomyocytes [23,24], have been confirmed in clinical research.

We consider that intracellular advanced glycation end-product (AGE)-induced cardiomyocyte dysfunction should be a novel non-ischemic type of CVD [25,26,27]. Certain AGE types, such as *N*^ε^-carboxymethyl-lysine (CML), *N*^ε^-carboxyethyl-lysine (CEL), *N*^δ^-(5-hydro-5-methyl-4-imidazolone-2-yl)-ornithine (methylglyoxal-derived hydroimidazolone) (MG-H1), glyoxal-derived hydroimidazolone (G-H1), dihydroxyimidazolidine, and 2-ammnonio-6-[4-(hydroxymethyl)-3-oxidopyridinium-1-yl]-hexanoate-lysine (4-hydroxymethyl-OP-lysine, hydroxymethyl-OP-lysine) have been associated with the generation and modification of proteins (RyR2, F-actin, and myosin) and cardiomyocyte dysfunction. [11,25,26,27,28].

Both ischemic and non-ischemic CVDs are associated with a lifestyle characterized by the excess consumption of saccharides (glucose and fructose) and lipids and should be categorized as lifestyle-related diseases (LSRDs) [29,30,31,32,33]. Excessive consumption of saccharides and lipids can result in obesity, diabetes mellitus (DM), CVD, and non-alcoholic steatohepatitis (NASH), which belong to LSRDs. Numerous researchers have warned that LSRDs are a serious issue in the 20th and 21st centuries and have since assessed the mechanisms by which excess nutrients facilitate and cause the progression of LSRDs [29,30,31,32,33]. 

Despite the seriousness of this human health issue, we believe that LSRDs have existed for thousands of years because ancient humans ate plants (*Parthenocissus tricuspidata (Siebold & Zucc.) Planch.* [Japanese name: Amazuru], *Gynostemma pentaphyllum (Thunb.) Makino* [Japanese name: Amachazuru]) that contain excess glucose and fructose, as well as mammalian meat (pork), which contains a high lipid ratio [34,35,36,37]. Ancient humans, such as royalty and nobility, likely obtained a high-nutrient diet similar to that of the 21st century. Additionally, we believe that ancient humans recognized the symptoms of certain LSRDs, such as DM [38,39,40,41], CVD [42], NASH [43,44], and hypertension [45], and treated them using traditional medicines worldwide, including East Asian medicines (e.g., Chinese, Korean, and Japanese) [38,39,40,41,42,43,44,45]. Preparations that originate from plants, animals, and minerals are used in traditional medicines. Numerous researchers have studied natural compounds in these medicines (e.g., quercetin, curcumin) to inhibit the generation of methylglyoxal-derived AGEs (MGO-AGEs) and their mechanisms, such as the carbonyl trap effect and glyoxalase-1 (GLO-1) activation, which remove/reduce methylglyoxal [46,47]. These findings may be crucial for understanding how traditional medicines prevent intracellular AGE-induced CVDs.

## 2. Regulatory Mechanisms of Contracting Cardiomyocytes

Cardiomyocytes require the appropriate circulation of intracellular Ca^2+^ to regulate beating [2,3,4,5,6]. This Ca^2+^ circulation is associated with SERCA2α and RyR2 in the SR. SERCA2α translocates Ca^2+^ into the SR, whereas RyR2 releases Ca^2+^ (Figure 1). In contrast, two filaments (thick and thin) associated with beating are present in the cytoplasm. The thick filament is primarily composed of myosin, and the thin filament is a complex of F-actin, tropomyosin, and troponin (F-actin–tropomyosin filaments) (Figure 1 and Figure 2) [7,8,9,10]. When cytoplasmic Ca^2+^ combines with troponin, it transmits a signal to the thick filament against F-actin in the thin filament to induce cardiomyocyte contraction. This contraction and relaxation occur rhythmically during cardiomyocyte beating. In contrast, Ca^2+^ accumulates in the mitochondria, indicating a lack of cytoplasmic Ca^2+^ [26]. 

## 3. CVD

### 3.1. Cardiomyocyte Dysfunction through Ischemia

Because cardiomyocytes require various materials (oxygen, glucose, and Ca^2+^) from the blood, ischemia is a serious obstacle to obtaining these nutrients (Figure 3) [12,13,14,15,16]. Atherosclerosis is caused by plaque accumulation, which narrows the luminal space where various materials flow. The major components of plaques are lipids (specifically cholesterol) and macrophages [12,13]. MMCS is induced by the osteoblastic transformation of vascular smooth muscle cells because hyperphosphatemia and secondary hyperparathyroidism play crucial roles in the pathogenesis of vascular calcifications [14,15]. Arteriocapillary sclerosis causes the occlusion and narrowing of arteries without plaque formation [16]. 

### 3.2. Cardiomyocyte Dysfunction through Non-Ischemia

The non-ischemic type of CVD can be induced by direct cardiomyocyte dysfunction (Figure 3) [17,18,19,20,21,22,23,24]. Examples include myocarditis [17,18], diabetic cardiomyopathy [19,20], cardiac tumors [21,22], injury by anticancer agents against cardiomyocytes [23,24], and intracellular AGEs. Additionally, intracellular AGE-induced cardiomyocyte dysfunction is a novel non-ischemic type of CVD (Figure 3) [25,26,27]. Intracellular MGO-AGE-modified proteins have been identified and quantified in cardiomyocytes from patients with heart failure, elucidating their relationship with dysfunction (detailed analysis is described in Section 6 below) [25,26,27]. 

## 4. AGEs

### 4.1. Various AGE Types

AGEs are generated by the non-enzymatic condensation of carbonyl groups in saccharides and their intermediate/side products (e.g., glyoxal, glycolaldehyde) and amino group residues in proteins (Figure 4) [48,49,50,51]. This is known as the Maillard reaction, first reported in 1912 [50]. However, saccharides and their intermediate/side products combined with amino acids (lysine [K] and arginine [R]) have also been named AGEs [48,49,50,51]. Although they share similarities with the low-molecular-weight structure of AGEs, numerous researchers have termed them as AGEs or free-type AGEs [48,49]. Therefore, we introduced free-type AGEs for AGE catalysis based on the original compounds. Furthermore, the routes of the generation of AGEs include the Maillard reaction, which requires the generation of Schiff bases and Amadori products, and other reactions (e.g., autooxidation) [49,50]. We introduced the generation routes of CML accordingly (Figure 4). CML can be generated from Amadori products, which originates from glucose (Figure 4). In contrast, glyoxal can be produced from the autooxidation of glucose, and CML is also able to be generated from it. Glycolaldehyde can be produced from Schiff bases and generate CML as well. Because glycolaldehyde is the origin of glyoxal, CML can also be generated from this route (Figure 4) [49,50].

Six categories of classical catalysis have been established as follows: glucose-derived AGEs (Glc-AGEs, AGE-1), glyceraldehyde-derived AGEs (GA-AGEs, AGE-2), glycolaldehyde-derived AGEs (AGE-3), MGO-AGEs (AGE-4), glyoxal-derived AGEs (AGE-5), and 3-deoxyglucosone-drived AGEs (3DG-AGEs, AGE-6) (Table 1, Figure 4 and Figure 5) [49,51]. Free-type AGEs have been identified using technologies, such as nuclear magnetic resonance (NMR) imaging and mass spectrometry (MS) (gas chromatography–MS [GC–MS], matrix-associated laser desorption/ionization–MS [MALDI–MS], and liquid chromatography–electrospray ionization–MS [LC–ESI–MS]) [48,52,53,54,55]. We have provided examples of AGEs, their origin, and categorization in Table 1. These data suggest that different origins may generate the same type of AGEs. However, we believe that the pathways are different and may affect the period for the generation of the same type of AGEs.

However, the identification of novel AGEs and evidence that similar AGEs were generated from compounds of various origins facilitated the revision of the classical categorization. In 2019, lactoaldehyde-derived AGEs [56] and glucose–lysine [57] were identified as novel AGEs. The former fits into the existing AGE categories. In 2021, novel GA-AGEs were reported, which were named pyrrolopyridinium lysine dimers derived from glyceraldehydes 1 and 2 (PPG1 and PPG2) [58]. In 2022, melibiose-derived AGEs (MAGE) were successfully identified and quantified in human plasma as a novel AGE, proposing MAGE as a novel AGE subgroup (AGE-10) [59]. CML has not been categorized as an MGO-AGE; however, the data have demonstrated that CML can be an MGO-AGE [28,60]. Recombinant thioredoxin can be treated with methylglyoxal to generate CML-modified thioredoxin [28]. CML can be generated from methylglyoxal using GC–MS [60]. Although CEL, MG-H1, argpyrimidine, and 6-{1-(5S)-5-ammnonio-6-oxido-6-oxyohexyl}-4-methyl-imidazolium-3-yl}-L-norleucine (MOLD) are the major MGO-AGEs (Figure 6) [49,50,61,62,63,64], CML may be categorized as an MGO-AGE [28,60]. In contrast, MG-H1 and argpyrimidine can be categorized as GA-AGEs because they are generated from glyceraldehyde (Figure 6) [65,66]. Though 3-hydroxy-5-hydroxymethyl-pyridinium (GLAP) [67,68], trihydoxy-triosidine [69], and PPGs [58] belong to GA-AGEs, there are no reports indicating that they are generated from methylglyoxal (Figure 6). The structure of a toxic AGE (TAGE), which is generated from glyceraldehyde, remains unclear, although a hypothesis has been proposed [70]. Based on these results, the concepts of MGO-AGEs and GA-AGEs can be enhanced (Figure 6). In contrast, methylglyoxal, glyoxal, glyceraldehyde, and glycolaldehyde can be generated via lipid oxidation [50]. They can also be derived without the metabolism of saccharides.

### 4.2. Crude, Diverse, and Multiple AGE Patterns

We focused on these various AGEs, modified proteins, and one free-type AGEs, which has more than two amino acid residues (e.g., MOLD) combined with over two proteins [50,54,70,71]. Therefore, we categorized crude AGE patterns (crude extracts containing AGE patterns; certain AGE types can be generated from one type of saccharide metabolites/derivative) (Figure 7), type 1 diverse AGE patterns (certain AGE structures can modify one type of protein, but not one molecular protein) (Type 1A; different types of AGE structures modified by the same amino acid residue in one type of protein; Type 1B: same types of AGE structures with different amino acid residue modifications in one type of protein; Type 1C: different types of AGE structures modified by different amino acid residues in one type of protein) (Figure 8a), type 2 diverse AGE patterns (one type of AGE structure modified by certain protein types) (Figure 8b), type 1 multiple AGE patterns (certain AGE type structures modified into one protein molecule, but not a specific type of protein) (Figure 9a), and type 2 multiple AGE patterns (the modification of the AGE structure involving more than two proteins through intermolecular covalent bonds) (Figure 9b) [71]. We believe the analysis of crude, diverse, and multiple AGE patterns is beneficial to reveal the function of AGE-modified proteins.

### 4.3. Intracellular AGEs and LSRDs

The intracellular AGEs, such as CML, CEL, MG-H1, argpyrimidine, GLAP, and TAGE were identified and quantified in vitro [11,26,72,73,74], in vivo [26,27,75,76], and in clinical studies [26,27]. An argpyrimidine-modified heat shock protein 27 (HSP27) was present in a gastric epithelial cell line (RGM-1) incubated in a normal glucose medium in vitro [72]. In contrast, a human pancreatic ductal carcinoma cell line (PANC-1) treated with glyceraldehyde, MG-H1, argpyrimidine, GLAP, and TAGE was generated in vitro [73,74]. CML and glucose–lysine, which were identified as novel AGEs, were found to be accumulated in the eye lenses of DM model rats and indicated that these AGEs can reduce visual activity in patients with DM [57]. CEL-modified proteins were generated in the skeletal muscles of C57/BI6j mice and leptin-deficient ob/ob mice fed with standard, high-fat, high-sugar, or 60% fructose diets [75,76]. MG-H1 and CEL-modified proteins were identified and quantified in patients with heart failure [26,27]. Despite the conclusions and hypotheses that each AGE induces cytotoxicity and facilitates various symptoms of LSRDs (DM, NASH, CVD, and sarcopenia) [11,26,72,73,74,75,76], we believe that crude, diverse, and multiple AGE patterns should be included in the analysis of cellular responses and LSRD symptoms [71]. 

### 4.4. Extracellular AGEs and LSRDs

#### 4.4.1. AGEs in the Body Fluids

Intracellular AGEs are generated and induce cytotoxicity, which can leak into bodily fluids (blood [59,77,78,79], saliva [79], and urine [80]). In contrast, the receptors for AGEs (RAGE) and toll-like receptor 4 (TLR4), which can bind to various AGE types, are expressed in almost all cells and organs (oral, esophagus, stomach, lung, liver, heart, and kidney) [81,82,83,84]. The AGEs–RAGE/TLR4 axis induces cytotoxicity, such as inflammatory reactions, and facilitates LSRDs [81,82,83,84]. Although AGEs in the body fluids are harmful, researchers have attempted to analyze the relationship between AGEs in body fluids and LSRDs because they may serve as beneficial biomarkers of LSRDs [59,77,78,79]. MAGE, a novel AGE, is a beneficial biomarker of NASH [59]. In contrast, AGEs with unclear structures can be used as biomarkers through antibody quantification [77]. CML, CEL, MG-H1, and *N*^ω^-carboxymethyl-arginine (CMA) were simultaneously quantified in the blood using ESI–MS analysis, which revealed that MG-H1 is associated with nephropathy [78]. An enzyme-linked immunosorbent assay (ELISA) was developed to quantify free pentosidine in urine, and its accuracy was compared with that of high-performance liquid chromatography (HPLC) [80].

#### 4.4.2. AGEs in the Extracellular Matrix

Collagen, which is a part of the extracellular matrix (ECM), located among the vascular endothelial cells [85], can be modified by AGEs. Intracellular collagen can also be modified by AGEs which are secreted from the cells and thus induce cellular dysfunction. 

#### 4.4.3. Dietary AGEs

Saccharides and proteins/peptides in beverages and foods undergo heating during manufacturing and cooking, resulting in the formation of AGEs, which are named dietary AGEs [86,87,88,89,90]. Dietary AGEs can combine with RAGE and TRL4 to induce cellular cytotoxicity and organ dysfunction. The relationship between CML and cancer risk has demonstrated that a low CML intake may reduce the risk of liver cancer [87]. Dietary AGEs may induce inflammation through the AGE–RAGE axis in the stomach [89]. CML, CEL, and MG-H1 were quantified along with the baseline intake of dietary AGEs from food [90]. Therefore, crude, diverse, and multiple AGE patterns can occur in dietary AGEs from beverages and foods and in intracellular AGEs [71]. Furthermore, the effects of dietary AGEs on the human body are influenced by digestion and absorption as they are introduced via the mouth [50]. Though the digestion of dietary AGEs may start in the oral epithelial cells, by salivary enzymes such as α-amylase, most of them are directed towards the gastric track and intestinal phase, where the absorbed AGEs are delivered to the circulatory system and any residual AGEs are then excreted in urine [50].

## 5. Types of Identification and Quantification Technologies for AGEs

### 5.1. Fluorimetry

AGE fluorescence was measured to quantify the AGEs [91]. This is beneficial for researchers quantifying AGEs where each of the structures or cannot be identified. However, other fluorescent substances may also affect this method. This characteristic of fluorimetry ensure that both free-type AGEs and AGEs-modified proteins can be quantified. AGEs are generally excited at a wavelength of 370 nm and when the emitted fluorescence is at 440 nm. Due to the general characteristics of this analysis, the types of .AGEs (e.g., MG-H1, argpyrimidine) are indistinguishable. However, Pinoto et al. reported that they managed to quantify pentosidine because it is excited at a wavelength of 370 nm and fluorescence emitted at 378 nm [91].

### 5.2. Immunostaining

Immunostaining analysis can identify and quantify AGEs in cells and tissues. However, each structure remains unclear because anti-AGE antibodies recognize all the AGE structures [92]. Various techniques that require anti-AGE antibodies, such as immunostaining [92], Western blotting [11,75,76,93], slot blotting [11,75,76,93], and ELISA [77,80,94], can be grouped together. The respective advantages of these approaches are that anti-AGEs antibodies can recognize and target AGEs, but the specific structures remain unclear. In contrast, the limitation of these analyses is their inability to define the modification site on the AGE-modified protein (Figure 8 and Figure 9). However, these technologies are suitable to analyze AGE-modified proteins (or AGE-modified peptides). Because the preparation of antibodies which can recognize and combine low-molecular compounds is difficult, it is believed that they are unsuitable for the analysis of free-type AGEs [48]. We previously detected a type of GA-AGE with an unclear structure in cardiomyocytes in vitro [11]. However, argpyrimidine and pentosidine were detected in the pancreatic islets of the obese New Zealand mice [92]. Because the location of intracellular AGEs is visible, immunostaining is useful for analyzing their association with cell and/or tissue dysfunction [11]. 

### 5.3. Western Blotting

Western blotting analysis can detect AGE-modified proteins transferred onto membranes (nitrocellulose and polyvinylidene difluoride [PVDF]) through chemiluminescence or fluorescence [11,75,76,93]. An advantage of this technique is that the molecular weight of each AGE-modified protein can be determined using protein molecular weight markers. We have previously reported on certain GA-AGE-modified proteins in PANC-1 cells using anti-GA-AGE antibodies, although the structure of these GA-AGEs has not been proven [48]. Additionally, we revealed high molecular weights of HSP 27, 70, and 90 βM, which may be modified using GA-AGEs, in the same study [48]. CML- and CEL-modified proteins were detected and quantified in murine skeletal muscles based on the detected bands [75,76]. We then defined the type 2 diverse AGE pattern as a type of AGE structure that can be modified for various proteins [71].

### 5.4. Slot Blotting

Although slot blotting requires a membrane onto which the proteins can be transferred, similar to that which is used for Western blotting, this method cannot analyze the molecular weight of each AGE-modified protein [11,48,74,94,95,96,97]. However, we believe that slot blotting is useful for quantifying the total AGE-modified proteins using anti-AGE antibodies [48,98]. Although numerous methods have used nitrocellulose membranes and general commercial buffers containing Triton-X, we considered PVDF membranes over nitrocellulose membranes for proteins [48]. However, Triton-X was unsuitable for this analysis because it may inhibit that proteins were probed onto the PVDF membrane; therefore, Dr. Takata selected a customized buffer comprising tris-(hydroxymethyl)-aminomethane (Tris), urea, thiourea, and 3-[3-(cholamidopropyl)-dimethylammonio]-1-propanesulfonate (CHAPS) and developed novel slot blotting in 2017 [74]. This buffer facilitates the probing of proteins (methylated, acetylated, phosphorylated, and AGE-modified proteins) on the PVDF membrane surface through carbamylation [48,98]. Studies have since quantified GA-AGEs using Dr. Takata’s lysis buffer and PVDF membranes [11,74,94,95,96,97].

### 5.5. ELISA

ELISA, which uses chemiluminescence and fluorescence as emission signals on antibody binding, employs a principle similar to that of slot blotting to identify and quantify AGEs [26,77,79,80,94]. However, the sample volume limitation (100–200 μL) of ELISA is lower than that of slot blotting because it is based on membrane chromatography [98]. In contrast, we believe that AGEs in body fluids, such as blood [77,94], saliva [79], and urine [80] can be more extensively quantified than that of slot blotting. Additionally, researchers can quantify intracellular AGEs in cell lysates when materials, such as anti-AGE antibodies, are suitable for ELISA [26]. 

### 5.6. GC–MS

GC analysis was developed in 1950 and coupled with MS technology to develop GC–MS [60,99,100]. GC–MS is used to quantify low-molecular-weight compounds (100–1000 Da). Therefore, AGE-modified proteins are unsuitable for analysis with GC–MS. In contrast, free-type AGEs are suitable for analysis with GC–MS, where researchers can identify and quantify the free-types AGEs which would have been prepared from acid hydrolysis of the AGE-modified proteins [60,99,100]. However, free-type AGEs need to undergo esterification to be prepared as samples for GC-MS because the sample must be highly volatile [48]. If the structure of the AGEs remains unclear (data on the structure of AGEs are not entered), researchers would be unable to detect the AGEs. Considering this characteristic of requiring data on the structure of free-type AGEs, GC– [60,99,100], ESI– [26,27,52,55,57,101], and MALDI–MS [53,54] can be grouped together. Researchers have quantified free-type AGEs (CML and CEL) using GC–MS; however, they should also (i) input the molecular data of free-type AGEs into the analysis module and (ii) prepare ester derivatives of the samples. Using GC–MS, researchers can perform the absolute quantification of free-type AGEs in the samples with standard free-type AGEs based on each ion peak in the analysis [60,99,100]. 

### 5.7. ESI–MS and MALDI–MS

ESI– and MALDI–MS have contributed to the identification of novel AGEs (free-type AGEs). ESI–MS equipment is generally connected to HPLC equipment to establish HPLC–ESI–MS [54,55,56,57,58,101]. NMR and HPLC–ESI–MS/MALDI–MS can identify the structure of free-type AGEs—the signals of protons, carbon, and nitrogen are detected through NMR, and the precursor and fragment ion peaks are detected through HPLC–ESI–MS/MALD-MS [54,55,56,57,58]. Once the researchers have succeeded in identifying the structure of free-type AGEs, they can identify them in the samples (cell lysates and serum) because data on the precursor and fragment ion peaks are available. Additionally, if standard free-type AGEs are obtained, the free-type AGEs in the samples can be identified based on both the whole and fragment peaks [48,71]. Free-type CML, CEL, MG-H1, and CMA can be quantified in the serum of patients using HPLC–ESI–MS [77]. Following acid hydrolysis of the AGE-modified proteins, the analysis of the free-type AGEs can be performed with ESI–/MALDI–MS as well as GC–MS [48]. Because esterification of the free-type AGEs does not need to be performed in ESI–/MALDI–MS, these technologies are more beneficial compared with GC–MS. In contrast, these technologies have been developed to identify and semi-quantify the AGE-modified peptides [26,27,71,72,73,102]. They can identify the amino acid residues responsible for AGE structural modifications [26,27,102]. These HPLC–ESI–MS/MALDI–MS technologies have facilitated the development of type 1 and 2 diverse and type 1 multiple patterns (Figure 8 and Figure 9a). The limitation of ESI– and MALDI–MS is that the automatic identification of both free-type AGEs and AGE-modified peptides is not possible if their structural data has not been entered into the database. Furthermore, we believe that type 2 multiple AGE patterns might be detected using automatic ESI– and MALDI–MS analysis because the analysis software is able to recognize a fragment ion peak in which the AGE peptide structures might be modified by a single amino acid residue (Figure 9b). [71]. 

### 5.8. NMR

The novel free-type AGEs, which were synthesized in tubes and isolated from samples such as plasm, could be identified with NMR. Analysis of the data of the signals of each element (e.g., proton, carbon, nitrogen) indicated their combinations (e.g., C-H, O-H, N-H) [55,56,57,58,59]. Once their structures had been identified, the data of the NMR could then be used to identify the ones which were isolated in samples. However, the detection of the AGE-modified proteins with NMR was difficult. Though the signal of each element appears to be proportional to the existence of the free-type AGEs, there have been no reports that the quantification of AGEs in samples can be performed with NMR.

## 6. Intracellular AGEs and Cardiomyocytic Dysfunction

### 6.1. Role of AGEs in the Pathophysiology of DM and Hypertension

The role of AGEs in the pathophysiology of DM and hypertension has been assessed [25]. In the study, cardiomyocytes were prepared from the left ventricles of patients who underwent coronary bypass grafting and divided into the following three groups: (i) control (coronary bypass grafting patients without DM and hypertension), (ii) hypertension, and (iii) DM and hypertension (DM + hypertension). The location of CML in the cardiomyocytes was analyzed in each group, and no significant differences in CML counts were observed among the three groups. Additionally, there was no significant correlation between the CML count and the body mass index (BMI) in the hypertension and DM + hypertension groups. In contrast, there was a positive correlation between the CML count and BMI in the control group. These results indicate that intracellular CML may not facilitate cardiomyocyte dysfunction despite the accumulation of intracellular CML. This may be associated with risk factors for LSRDs. However, we focused on the generation and accumulation of intracellular CML in cardiomyocytes. Although previous studies have indicated that intracellular CML may not induce dysfunction in cardiomyocytes, LeWinter et al. analyzed only the intracellular CML, but not other types of AGEs. We believe that crude, diverse, and multiple AGE patterns may be useful for analyzing LSRDs, such as CVD (Figure 7, Figure 8 and Figure 9). In this investigation, the intracellular role of CML in cardiomyocytes remains unclear. However, we can provide useful data to predict their functional activity. Mastrocola et al. reported that intracellular CML-modified proteins were generated/accumulated in skeletal muscle in C57B1/6J mice fed a high-fructose diet, and that they may induce lipogenesis [76]. CML-modified proteins might induce lipogenesis in cardiomyocytes. If the localization/accumulation of other types of AGEs and cardiomyocyte dysfunction were analyzed, the novel combined effects of CML and these modified proteins on cardiomyocytic dysfunction may be discovered.

### 6.2. Various Glycations in RyR2 in the Cardiomyocytes in DM

Various glycations occur in the ryanodine receptor 2 (RyR2) in rat cardiomyocytes [103]. The rats were divided into the following three groups: (i) control, (ii) streptozotocin (STZ)-induced diabetes model (T1DM model), and (ii) STZ-induced diabetes model treated with insulin treatment (T1DM–insulin model). In this study, the protein levels of RyR2 in the T1DM model were lower than those in the other two groups. In contrast, the glycation levels of RyR2 in the T1DM model were higher than those in the other two groups. CML, 3-deoxyglucosone, argpyrimidine, pentosidine, pyraline, and GA-pyridine were detected in the RyR2 (Figure 4, Figure 5, Figure 6 and Figure 10). Although 3-deoxyglucosone is an intermediate of saccharides but not AGEs, the other compounds are AGEs. Although these glycations were confirmed through the immunoprecipitation of RyR2 and detection with specific anti-AGE antibodies, the results indicate that type 1 (A and B) AGE patterns may occur in RyR2 (Figure 8a). Additionally, RyR2 was transfected into the human embryonic kidney 293 (HEK-293) cell line, which was subsequently treated with methylglyoxal. The release of Ca^2+^ from glycated RyR2 was facilitated [103].

### 6.3. Intracellular AGEs Contain Glycated RyR2 in the Cardiomyocytes of the Senescence Model

The intracellular AGEs in the cardiomyocytes of C57BL/6 mice and surgical patients were analyzed [26]. In the animal experiment, two groups of C57BL/6 mice were prepared as follows: the (i) young (5–6 months) and (ii) old (≥20 months) groups. Each cardiac ventricle was minced and homogenized with lysis buffer to prepare the cardiomyocyte lysates. Intracellular CML-modified proteins (approximately 25–75 kDa) in the old group were significantly increased compared to those in the young group, as determined through Western blotting. In contrast, the heart tissue was homogenized, and the cell lysate was analyzed using ESI–MS. In the ESI–MS analysis, the hydroxymethyl OP-lysine adduct [104,105], MG-H1, dihydroxy-imidazolidine [106,107,108], G-H1 [109,110], *p*-hydroxyphenylglyoxal-arginine adduct [111,112], malonaldehyde-lysine adduct [113], and CML were selected as the glycation conditions for protein modification (Figure 5 and Figure 11). In the previous AGE category, hydroxymethyl OP-lysine, dihydroxy-imidazolidine, and G-H1 are glycolaldehyde-derived AEGs, MGO-AGEs, and GO-AGEs were selected, respectively [104,106,109]. Ruiz-Meana et al. further analyzed the protein levels of glyoxalase-1 (GLO-1), which metabolizes methylglyoxal to S-lactocylglutathione, and observed no significant differences between the two groups. Because GLO-1 can metabolize methylglyoxal and glyoxal, the activated GLO-1 contributes to their reduction and inhibits the generation of MGO-AGEs and GO-AGEs [104,106,109]. However, S-lactocylglultathione and D-lactate levels in cardiomyocytes were significantly reduced in the old group compared to those in the young group. These results indicated that both GLO-1 and GLO-2, which metabolize S-lactocylgultathione to D-lactate, were suppressed [26,101]. We hypothesized that intracellular MGO-AGEs (MG-H1, dihydroxy-imidazolidine, and CML) and GO-AGEs (G-H1) may increase because of the suppression of GLO-1 activity. These glycated proteins were identified, and their associations were analyzed. Although numerous glycated proteins belong to the mitochondrial, myofibril, and extracellular regions, Ruis-Meana et al. focused on glycated RyR2. The modification of RyR2 by MGO-AGEs was confirmed using anti-MGO-AGE (Cat. STA-011; Cell Biolabs, Inc., San Diego, CA, USA) and anti-RyR2 (Cat. ab196355; Abcam, Cambridge, UK) antibodies in immunostaining analysis. Because the locations of both the MGO-AGEs and RyR2 matched, the modification of RyR2 was revealed. In contrast, various AGE-modified peptides that were increased in RyR2 in the old group through HPLC–ESI–MS analysis, specifically detected increased levels of two peptides (EEKAKDEK [4448–4455] and REKEVAR [4601–4607]), which were modified by hydroxymethyl OP (Figure 11). Hydroxymethyl OP is generated from glycolaldehyde, but not methylglyoxal, and its increase is not associated with the suppression of GLO-1 [104]. Crude and type 1 diverse AGE patterns can be observed in this study (Figure 7 and Figure 8). The structure of MGO-AGEs remains unclear because the anti-MGO-AGE antibody can only recognize certain types of MGO-AGEs. Additionally, hydroxymethyl OP-modified proteins in RyR2 may be generated and accumulated because of senescence in cardiomyocytes. Moreover, Ruiz-Meana et al. revealed that Ca^2+^ accumulates in the interfibrillar mitochondria, but not in the subsarcolemmal mitochondria that are in close contact with the SR in cardiomyocytes. They reported the Ca^2+^ retention capacity in the interfibrillar mitochondria of aged mouse cardiac tissues. These results indicated that MGO- and hydroxymethyl OP-modified RyR2 may induce cardiomyocyte dysfunction (Figure 12). In contrast, the surgical patients were classified into two groups as follows: young (<75 years) and older (≥75 years) adults. Fresh myocardium was obtained and immediately processed to isolate the mitochondria and SR. The CML-modified proteins in the cardiomyocytes of the older aged group were significantly increased compared with those in the young group. Additionally, S-lactoylglutathione and D-lactate levels were significantly reduced in cardiomyocytes of the elderly group compared to those of the young group. The protein levels of GLO-1 exhibited no significant differences between the two groups. These results in the clinical studies align with those of the mice experiments. However, the total amount of MGO-AGEs in RyR2, which was analyzed using the biotinylated anti-MGO-AGE (Cat. HM5014; Hycult Biotech Inc., Wayne, PA, USA) and anti-RyR2 (Cat. ab196355; Abcam) antibodies through ELISA did not exhibit a significant difference. This may be because of species differences between humans and mice. In contrast, we should understand that malondialdehyde can also be generated via peroxidation of polyunsaturated fatty acids, and the malonaldehyde-lysine adduct can be accumulated from lipid metabolism [113] since CEL, MG-H1, dihydroxy imidazoline, and *p*-hydroxyphenyylglyoxal-arginine adduct were generated from methylglyoxal and glyoxal, and their accumulation might be associated with lipid oxidation [50,108,109,110,111,112].

### 6.4. Intracellular AGEs Contain Glycated Myosin and F-actin in Cardiomyocytes in Diabetes and Heart Failure

The left ventricles of donors were prepared from patients with (i) diabetes and heart failure (dbHF), (ii) donor hearts rejected for transplantation (NF), and (iii) dilated cardiomyopathy [27]. The cardiomyocyte lysates from the three groups were analyzed using dot blotting, which is similar to slot blotting and HPLC–ESI–MS. Dot blotting revealed that the MGO-AGE levels in the dbHF group were higher than those of the NF group. For ESI–MS analysis, which was performed in the MS/MS mode, the research conditions were set against MG-H1 and CEL. They identified 23 and 32 MGO-AGE-modified proteins in the NF and dbHF samples, respectively. The MGO-AGE-modified proteins, specifically actin and myosin, were identified and quantified. In the patient samples, CEL modification in actin (K70), α-myosin (K384), and skeletal myosin (K1899) in the dbHF group was significantly increased compared to that of the NF group. In contrast, CEL modification in actin (K293) and myosin (K180), and MG-H1 modification in α-myosin (R370) and the myosin essential light chain (R103) of murine skimmed cardiomyocytes (C57BL/6J, 3–4 months old) treated with methylglyoxal, were significantly increased compared to those of the control. Subsequently, the maximal calcium-activated force (F_max_) and calcium sensitivity in mouse-skimmed cardiomyocytes and human myocardial samples were analyzed. In mouse cardiomyocytes, methylglyoxal treatment delayed the fitted curve of force (mN/mm^2^) at each concentration of Ca^2+^ and suppressed F_max_ (mN/mm^2^), although Ca^2+^ sensitivity (EC_50_) increased. 

In contrast, dbHF-treated cardiomyocytes were also treated with methylglyoxal, and the fitted curve of the normalized force was analyzed for each Ca^2+^concentration. Methylglyoxal suppressed the fitted curve. Cardiomyocytes were obtained from the left ventricular tissue of patients with and without DM (these tissues were excised for transplantation from patients without HF) and analyzed for the glycation of myosin and actin using ESI-MS [114]. In patients with and without DM, 22 and 33 glycated myosin residues were detected, respectively. In the DM group, 14 glycated residues (K66, R237, R708, K740, R952, K1194, K1305, K1444, R1447, K1617, K1727, K1728, K1757, and K1771) were detected. In contrast, four and eight glycated residues were detected in patients with and without DM, respectively. In the DM group, five glycated residues (K70, K115, K118, R179, and R256) were detected [114]. Methylglyoxal, MG-H1, glyoxal, and G-H1 modified these residues in myosin and actin (Figure 4, Figure 5 and Figure 11) [114]. The results from two assessments by Papadaki et al. indicated that crude AGE and type 1 (A, B, C) and 2 diverse AGE patterns occurred, although the type 1 multiple AGE patterns were not detected (Figure 7 and Figure 8a) [27,71,114]. Papadaki et al. indicated the possibility that the glycation of myofilament proteins causes sarcomere dysfunction because the glycated myofilament reduces the Ca^2+^ sensitivity and calcium-activated forces in cardiomyocytes of patients with DM (Figure 13) [114]. However, the physiological effects (e.g., the number of beats per minute, force of the beats) in cardiac tissue have not been shown. Although they reported on the reduction of Ca^2+^ sensitivity and calcium-activated forces in the skimmed cardiomyocytes, which were treated with methylglyoxal in vitro, they did not identify and quantify the AGE-induced modifications [114]. The relationships between AGE-induced modifications in cardiomyocytes in the patients and the physiological effects should be researched in future.

### 6.5. Generation of TAGE in Cardiomyocytes

TAGE was generated in the glyceraldehyde-treated cardiomyocytes in vitro. Intracellular TAGE generation was detected and quantified using immunostaining and slot blotting [11]. In these cardiomyocytes, TAGE may reduce the beating rate and ratio of microtubule-associated protein light chain 3 (LC3)-II/LC3-I compared to that of the control. Because of this phenomenon [115], these results indicate that intracellular TAGE may suppress autophagy and reduce beating. However, treatment with glyceraldehyde-generated MG-H1, GLAP, and argpyrimidine-modified proteins may induce similar effects, although only TAGE was analyzed in this study [71,73]. 

Other types of AGEs, including TAGE, may cause this phenomenon. However, the mechanisms by which the suppression of autophagy inhibits beating remain unclear. Although the analysis was performed, we hypothesize that phospholamban was associated with a reduction in cardiomyocyte beating. Phospholamban, a low-molecular-weight protein (approximately 15 kDa), is located on SERCA2α and regulates Ca^2+^ uptake [116]. Phospholamban levels are regulated by autophagy [116]. If autophagy is suppressed by AGEs, beating dysfunction may be induced by the amount of phospholamban present. However, this hypothesis has not been proven. The experiments by Takata et al. were performed in vitro; however, it remains unclear whether these phenomena occur in cardiomyocytes in animal models or in clinical studies.

### 6.6. Highlights of the Novel Aspects of Our Research Compared to the Previous Investigations

In previous investigations (e.g., immunostaining, Western blotting, slot blotting, and ELISA), researchers have revealed the existence of individual AGEs and the AGEs-modification of protein (e.g., RyR2) using anti-AGEs antibody. These analyses were able to define a crude AGE pattern (Figure 7). Furthermore, various types of AGE-modified proteins with different patterns (Type 1 and 2 diverse patterns, and type 1 multiple AGE pattern) were identified (Figure 8 and Figure 9b). Each AGE structure may induce dysfunction in combination, or one AGE structure may prevent dysfunction while another AGE structure induces dysfunction. Based on the development of the technology for the analysis of AGE structures and their modifications, additional research is required to identify individual AGE-modified proteins in cardiomyocytes and to reveal the mechanisms of intracellular AGEs induction of CVD.

### 6.7. The Possibility That Extracellular AGEs Directly Induce Dysfunction of Cardiomyocytes

Extracellular AGEs, which leak into the blood, and dietary AGEs induce cardiomyocyte dysfunction in vitro [117,118,119]. However, the mechanisms by which these AGEs directly induce cardiomyocyte dysfunction in heart failure remain unclear in clinical studies and animal models. Because this research aims to prove these mechanisms, we could not avoid discussing the potential for extracellular AGEs to directly induce heart failure.

## 7. Inhibition of the Generation of AGEs by Natural Compounds in Traditional Medicines

### 7.1. Traditional Medicines

Throughout human history, various traditional medicines have been practiced [48]. They have been used extensively worldwide, including in Europe, North and South America, Asia, Africa, and Australia. Moreover, the plants and their places of origin within each area have since been determined, and their natural phytochemical compositions have been assessed in terms of their potential to prevent or treat various diseases [47]. In East Asia, ancient Chinese medicine has a potent influence on the surrounding areas (Korea and Japan), and Japanese traditional medicine (Kampo) has been developed based on Chinese traditional medicine [120,121,122,123]. Therefore, Kampo and traditional Chinese medicines are common [124,125,126,127]. In contrast, the recent Kampo (a narrow sense Kampo) medicines that are recorded in the Japanese pharmacopoeia are manufactured products regulated by Japanese law and are available in hospitals and pharmacies [128,129,130,131]. Although Kampo medicines are provided as granules, powders, capsules, or tablets, the original Kampo medicines contained traditional medicines derived from various plant parts, animals, or minerals. The mechanisms by which mokuboito (Chinese name: Mu-Fang-Ji-Tang), which contains four traditional medicines, protects cardiomyocytes by affecting the angiotensin II receptor have been demonstrated [132].

### 7.2. Various Natural Compounds Have Been Identified in Traditional Medicines

Numerous researchers have attempted to reveal the mechanisms of action of traditional medicines, such as Kampo medicines and traditional medicines [133,134]. However, this analysis is challenging because (i) several constituents are present in traditional medicines [135,136,137,138] and (ii) traditional medicines are orally administered, where their compounds are then digested, absorbed, and metabolized [139,140]. Therefore, traditional medicines have been administered to model animals and their effects have been analyzed [141,142]. This approach can demonstrate the effects of individual compounds, although the mechanisms of action of traditional medicines require further comprehensive analysis. Therefore, single phytochemicals have been used in vitro to treat cells or administered to animals to analyze their effects and reveal how individual compounds may affect various cells and tissues. Experiments with the cultured cells of some organs (e.g., heart, kidney, brain, lung, skeletal muscle) should be treated with the low-molecular-weight compounds in vitro and aimed at model the effects of the final metabolized compounds on cells. 

### 7.3. Inhibition of the Generation of Intracellular AGEs

In the cell, the mechanisms by which intracellular AGEs accumulate include the following: (i) inhibition of the generation of AGEs [143,144,145] and (ii) facilitation of AGEs clearance (autophagy and ubiquitin–proteasome system) [146]. Numerous researchers have focused on AGE inhibition because their mechanisms are simple to analyze. The primary method for the inhibition of the various AGEs is by using a carbonyl trap, in which a low-molecular-weight compound (e.g., aminoguanidine) combines with the intermediate/non-enzymic products from saccharides (methylglyoxal, glyoxal, and glyceraldehyde) [143]. The precursors of AGEs contain aldehyde or ketone groups (both classified as carbonyls), which can react with the amino acid residue (e.g., lysine, arginine) to generate AGEs. Compounds that can react with aldehyde or ketone groups of AGE precursors inhibit the generation of AGEs [143]. This is a very simple method to inhibit the generation of AGEs; however, such inhibitors (e.g., aminoguanidine) may induce cytotoxicity. For AGEs generated from methylglyoxal and glyoxal, the activation of GLO-1 has been the focus of research because it can eliminate methylglyoxal and glyoxal [144,145]. Although aminoguanidine and a GLO-1 activator, which are non-natural products, have been used to inhibit AGE generation, these reagents are cytotoxic and are not used in the clinical stage [143,144,145]. Low-molecular-weight natural compounds in traditional medicines inhibit the generation of intracellular MGO-AGEs and GO-AGEs (MG-H1, argpyrimidine, CML, CEL, and G-H1) through the carbonyl trap system and/or GLO-1 activation [46,47]. Quercetin [147], chrysin [148], genistein [149], epigallocatechin-3-gallate [150], *p*-hydroxy cinnamic acid [151], (+)-catechin [152,153,154], (-)-epicatechin [152,153,154], aspalathin [155], and hesperidin [156] inhibited AGE generation based on the carbonyl trap system (Figure 14) [46,47]. In contrast, imperialin [157], piperine [158], and dipheloretobohydroxycarmalol [159] activated GLO-1 to inhibit the generation of MGO-AGEs and GO-AGEs (Figure 14). Resveratrol [158,160,161,162] and curcumin [158,163] inhibited the generation of MGO-AGEs and GO-AGEs using both carbonyl trap system and GLO-1 activation (Figure 14) [158,160,161,162]. Because these effects of low-molecular-weight natural compounds were revealed in an in vitro assessment, the carbonyl trap system and GLO-1 activation will occur in cardiomyocytes if these compounds can be ingested. Although the general natural compounds in traditional medicines are orally administered, digested, absorbed, and metabolized before entering the blood and cardiomyocytes, the low-molecular-weight natural compounds illustrated in Figure 14 may not exhibit the effects achieved by oral administration of traditional medicines. However, they may inhibit the generation of AGEs, such as MGO- and GO-AGEs, through intravenous injection. Quercetin [147], chrysis [148], genistein [149], epigallocatechin-3-garate [150], (+)-catechin [152,153,154], (-)-epicatechin [152,153,154], aspalathin [155], and hesperidin [156] are flavonoids. In a meta-analysis of cohort studies, the intake of flavonoids prevented the risk of CVD. These intracellular AGEs may be involved in enhancing the dysfunction of cardiomyocytes within the context of different types of CVD and are beneficial for our investigation [164,165,166]. Additionally, the discovery of these natural compounds may result in an analysis of the mechanisms by which traditional medicines cure heart failure through intracellular AGEs in cardiomyocytes.

## 8. Conclusions

Various intracellular AGE-modified proteins (RyR2, F-actin, and myosin) have been identified and quantified in cardiomyocytes. Based on the results of this review, AGE generation can be categorized as follows: (i) crude AGE patterns, (ii) type 1 (1A, 1B, 1C) AGE patterns, and (iii) type 2 AGE patterns. The mechanisms underlying the dysfunction induced by AGE-modified RyR2, F-actin, and myosin have been elucidated. The future perspectives of this work include the following: (i) the identification of low-molecular-weight compounds in traditional medicines that are able to inhibit the generation of intracellular AGEs in cardiomyocytes and prevent CVDs and (ii) the identification of the underlying mechanisms of these compounds in their prevention of CVDs.

## Figures and Tables

**Figure 1 ijms-25-07319-f001:**
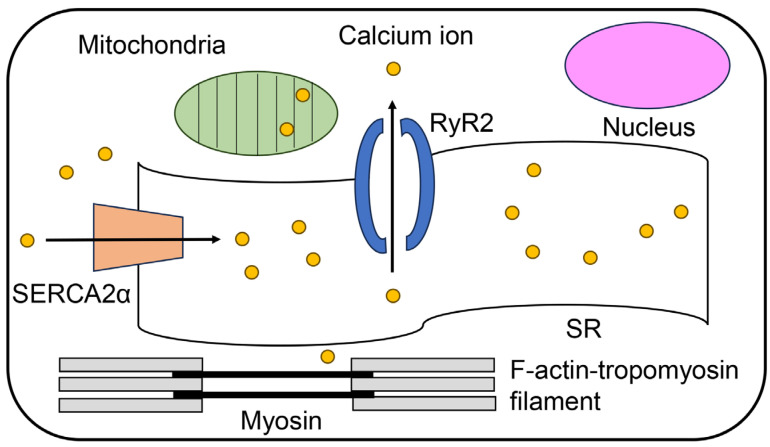
Model image of a cardiomyocyte [2,3,4,5,6]. Closed pink and green circles indicate the nucleus and mitochondria, respectively. The open ribbon indicates the cytoplasmic reticulum, and the closed orange trapezoid indicates sarco/endoplasmic reticulum Ca^2+^ ATPase 2α (SERCA2α). The closed blue hemicircle indicates ryanodine receptor 2 (RyR2) and yellow circles indicate calcium ions. The closed gray and black squares indicate myosin and F-actin–tropomyosin filaments (F-actin, tropomyosin, and troponin complex), respectively.

**Figure 2 ijms-25-07319-f002:**
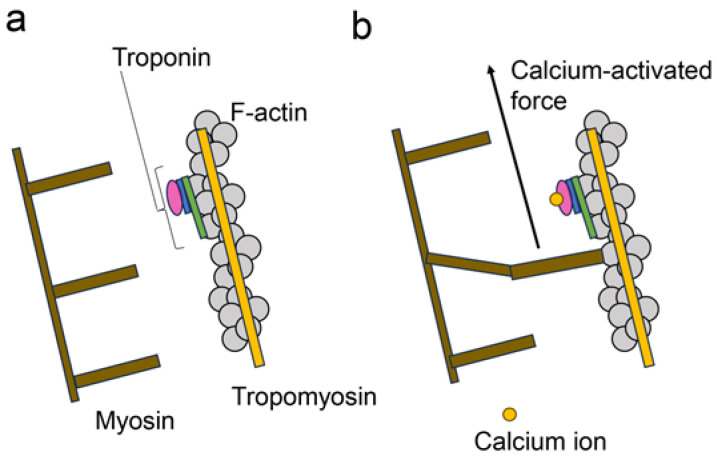
Myosin, F-actin–tropomyosin filament, and calcium-activated force [7,8,9,10]. The open brown squares indicate myosin, and the closed gray circle indicates F-actin. Yellow and green rectangles indicate tropomyosin and troponin T, respectively. The blue rectangle indicates troponin I, and the pink circle without characters indicates troponin C. The closed yellow circle indicates calcium ions. The black arrow indicates calcium-activated force. (**a**) The calcium ions do not combine with troponin. (**b**) The calcium ion combines with troponin C, and myosin moves the F-actin–tropomyosin filament.

**Figure 3 ijms-25-07319-f003:**
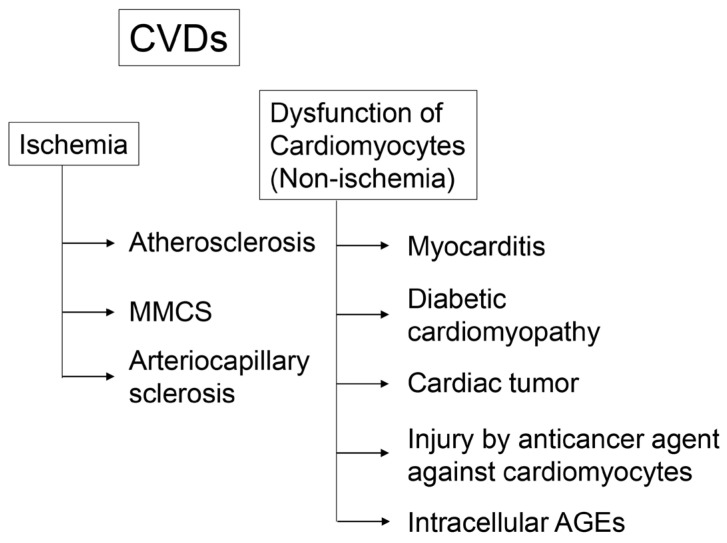
The pattern of cardiovascular diseases (CVDs). CVD; atherosclerosis [12,13]; Mönckeberg’s medial calcific sclerosis (MMCS) [14,15]; and arteriocapillary sclerosis [16]. Myocarditis [17,18]; diabetic cardiomyopathy [19,20]; cardiac tumor [21,22]; injury by anticancer agents against cardiomyocytes [23,24]; and intracellular advanced glycation end-products (AGEs).

**Figure 4 ijms-25-07319-f004:**
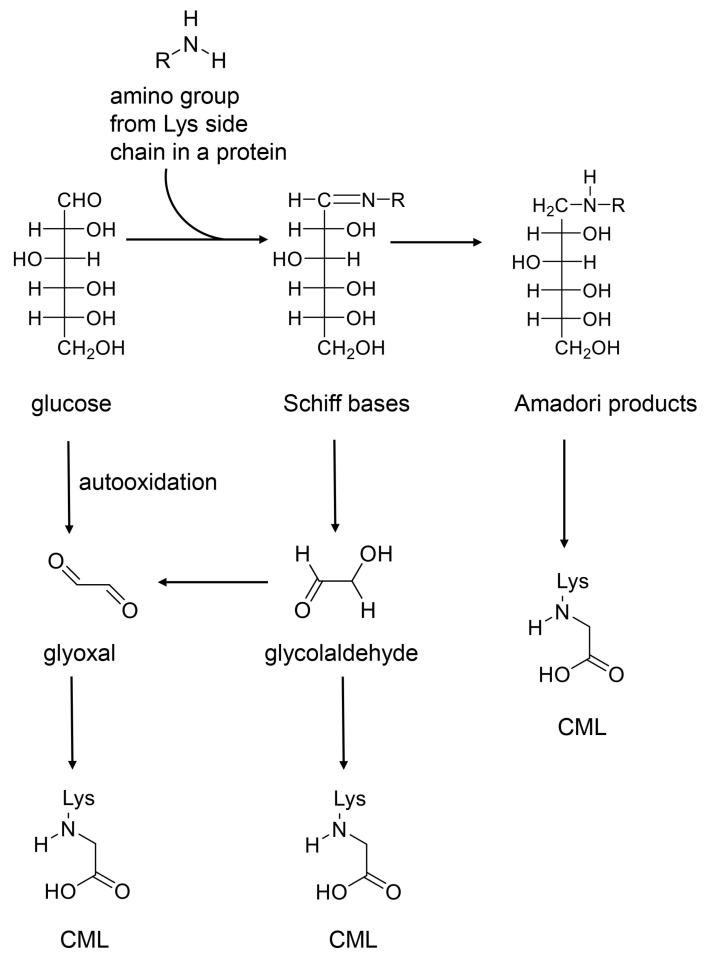
The routes of the generation of CML. R; proteins. Lys; lysine. CML, *N*^ε^-carboxymethyl-lysine. CML is generated from glucose through the Maillard reaction. In contrast, CML can be generated from glyoxal and glycolaldehyde, which are produced from glucose, though the route does not mediate the production of Amadori products [49,50].

**Figure 5 ijms-25-07319-f005:**
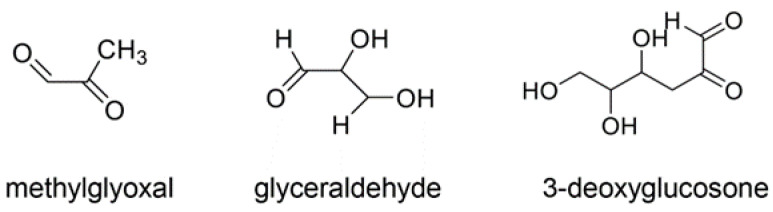
The intermediate/side products of saccharides. Each carbonyl is the origin of methylglyoxal-, glyceraldehyde-, and 3-deoxyglucosone-derived AGEs (AGE-4, -2, and -6), respectively [49,50].

**Figure 6 ijms-25-07319-f006:**
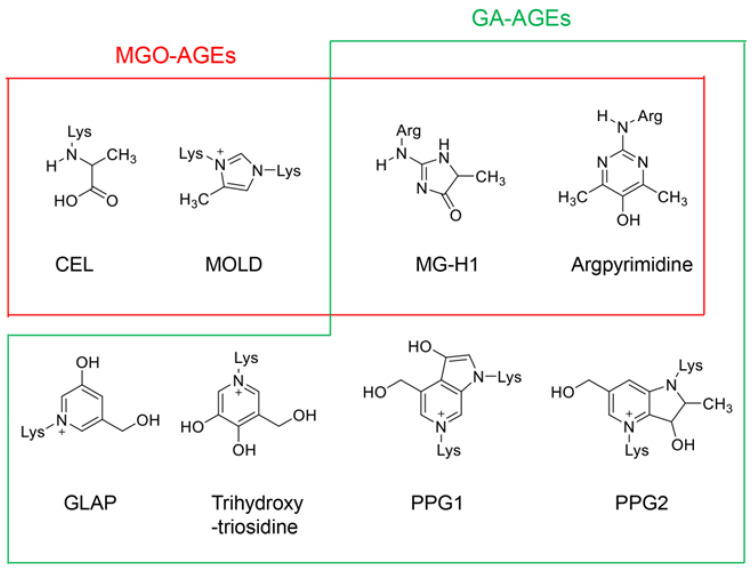
Each structure of the MGO-AGEs and GA-AGEs. MGO-AGEs, methylglyoxal-derived advanced glycation end products [49,51]; GA-AGEs, glyceraldehyde-derived AGEs [49,51]; Lys; lysine residue; Arg; arginine residue; The open red box indicates the classical MGO-AGEs. The open green box indicates the GA-AGEs. CEL, *N*^ε^-carboxyethyl-lysine [50,60]; MOLD, 6-{1-(5S)-5-ammnonio-6-oxido-6-oxyohexyl}-4-methyl-imidazolium-3-yl}-L-norleucine [50,54]; MG-H1, *N*^δ^-(5-hydro-5-methyl-4-imidazolone-2-yl)-ornithine (methylglyoxal-derived hydroimidazolone). MG-H1 can be generated from both methylglyoxal and glyceraldehyde [50,61,62,65]; argpyrimidine can be generated from both methylglyoxal and glyceraldehyde [50,63,66]; GLAP, 3-hydroxy-5-hydroxymethyl-pyridinium [67,68]; trihydroxy-triosidine [69]; PPG 1 and 2, pyrrolopyridinium-lysine dimer derived from glyceraldehyde 1 and 2 [58].

**Figure 7 ijms-25-07319-f007:**
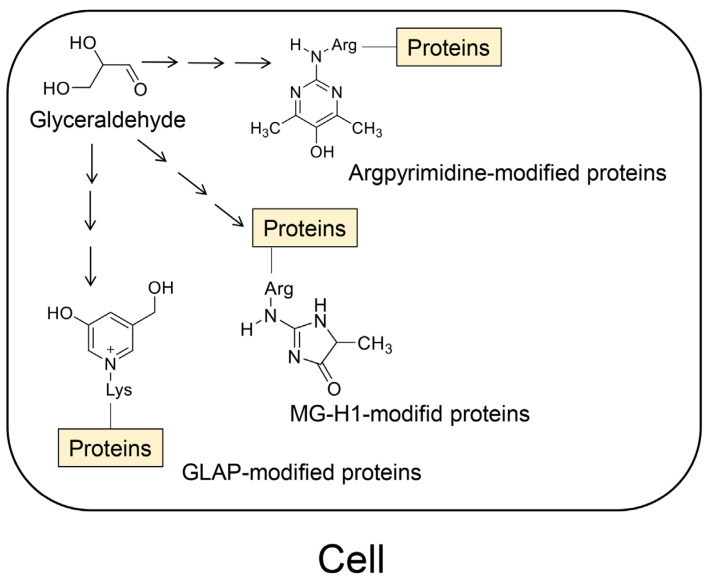
Crude advanced glycation end-product (AGE) pattern in the cell [71]. AGE; GLAP-, MG-H1-, and argpyrimidine-modified proteins belong to glyceraldehyde-derived advanced glycation end-products (GA-AGEs). GLAP, 3-hydroxy-5-hydroxymethyl-pyridinium; MG-H1, *N*^δ^-(5-hydro-5-methyl-4-imidazolone-2-yl)-ornithine (methylglyoxal-derived hydroimidazolone). Black arrows indicate several steps.

**Figure 8 ijms-25-07319-f008:**
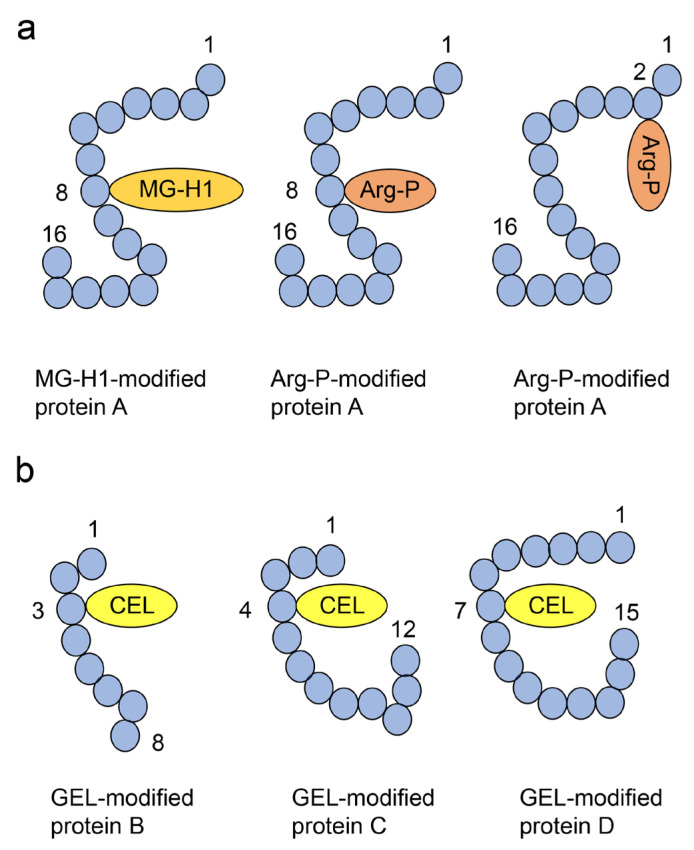
Type 1 (1A, 1B, 1C) and 2 diverse advanced glycation end-product (AGE) pattern [71]. The blue circles indicate the amino acids. Numbers represent the number of amino acid residues in proteins A, B, C, and D. MG-H1, *N*^δ^-(5-hydro-5-methyl-4-imidazolone-2-yl)-ornithine (methylglyoxal-derived hydroimidazolone); Arg-P, argpyrimidine; CEL, *N*^ε^-carboxyethyl-lysine. (**a**) Type 1 diverse AGE pattern. Each protein A is modified by a certain AGE structure, respectively. Based on the left and middle glycated protein A, we indicate the type 1A diverse AGE pattern (each protein A is modified by MG-H1 and Arp-P in the eighth amino acid residue, respectively). Based on the middle and right glycated protein A, we indicate the type 1B diverse AGE pattern (each protein A is modified by Arg-P at the eighth and second amino acid residue, respectively). Based on the left and right glycated protein A, we indicate the type 1C diverse AGE pattern (each protein A is modified by MG-H1 and Arg-P in the eighth and second amino acid residue, respectively). (**b**) Type 2 diverse AGE pattern. CEL modifies proteins B, C, and D, respectively.

**Figure 9 ijms-25-07319-f009:**
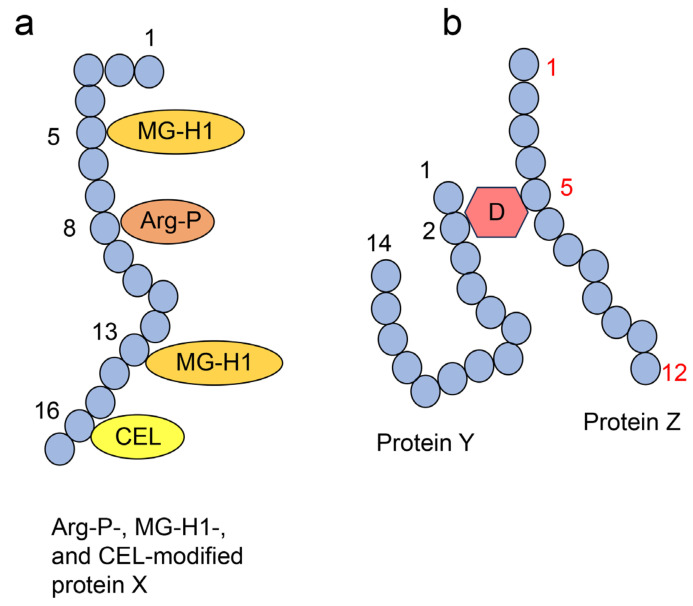
Type 1 and 2 multiple advanced glycation end-product (AGE) patterns [71]. The blue circles indicate the amino acids. (**a**) Type 1 multiple AGE pattern. Protein X (one molecular, but not one type of protein) is modified by MG-H1, Arg-P, and CEL. The black numbers represent the number of amino acid residues in protein X. MG-H1, *N*^δ^-(5-hydro-5-methyl-4-imidazolone-2-yl)-ornithine (methylglyoxal-derived hydroimidazolone); Arg-P, argpyrimidine; and CEL, *N*^ε^-carboxyethyl-lysine. (**b**) Type 2 multiple AGE pattern. The black and red numbers represent the number of amino acid residues in proteins Y and Z, respectively. D1: AGE structure that can combine between the second amino acid residue in protein Y and the fifth amino acid residue in protein Z.

**Figure 10 ijms-25-07319-f010:**
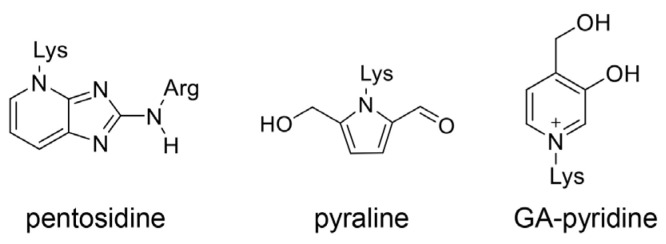
Advanced glycation end products (AGEs) are modified in ryanodine receptor 2 (RyR2) [103]. Pentosidine [50,54], pyralline [50,54], and GA-pyridine [50,54,103].

**Figure 11 ijms-25-07319-f011:**
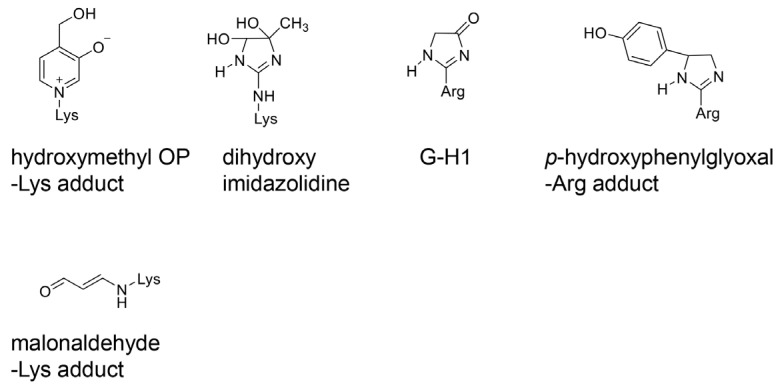
Advanced glycation end-products (AGEs) analyzed using electrospray ionization–mass spectrometry (ESI–MS) in the study by Ruiz-Meana et al. [26]. Lys, lysine; Arg, arginine; Hydroxymethyl OP-Lys [104,105], dihydroxyimidazolidine [106,107,108], G-H1 [109,110], *p*-hydroxyphenylgloxal-Arg adduct [111,112], and malonaldehyde-Lys adduct [113].

**Figure 12 ijms-25-07319-f012:**
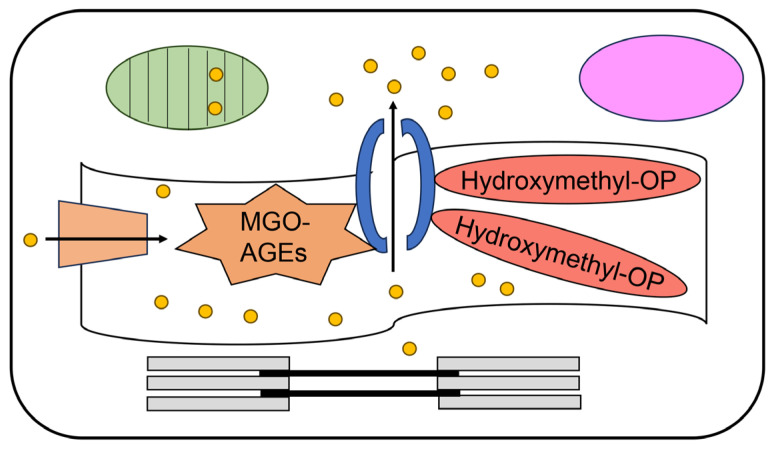
Hydroxymethyl-OP modified RyR2 in cardiomyocytes. The activity of RyR2 decreased, and the leak of calcium ion increased. Green circle indicates mitochondria. Open ribbon indicates the sarcoplasmic reticulum. Orange trapezoid indicates SERCA2α. Open blue circle indicates RyR2. Yellow circles indicate calcium ions. Gray rectangles indicate myosin. Black rectangles indicate F-actin-tropomyosin filament. MGO-AGEs, methylglyoxal-derived AGEs. Hydroxymethyl-OP,2-ammnonio-6-[4-(hydroxymetyl)-3-oxidopyridinium-1-yl]-hexanoate-lysine [104,105].

**Figure 13 ijms-25-07319-f013:**
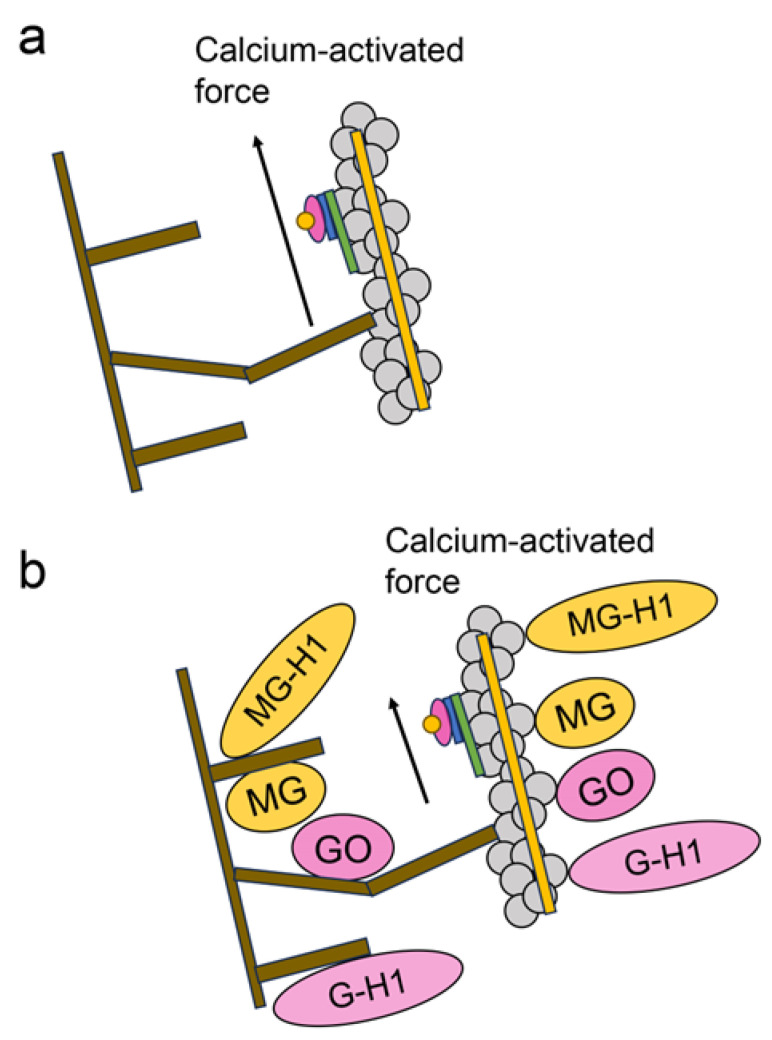
The predicted model that glycated myosin and F-actin–tropomyosin filament, and calcium-activated force [114]. The open brown squares indicate myosin, and the gray circles indicate F-actin. Yellow and green rectangles indicate tropomyosin and troponin T, respectively. The blue rectangles indicate troponin I and the pink circles with no text indicate troponin C. The yellow circles indicate calcium ions. The black arrow indicates the calcium-activated force. (**a**) Normal myosin and actin–tropomyosin filaments. (**b**) Glycated myosin and actin–tropomyosin filaments. MG, methylglyoxal; MG-H1, *N*^δ^-(5-hydro-5-methyl-4-imidazolone-2-yl)-ornithine (methylglyoxal-derived hydroimidazolone); GO, glyoxal; and G-H1, glyoxal-derived hydroimidazolone.

**Figure 14 ijms-25-07319-f014:**
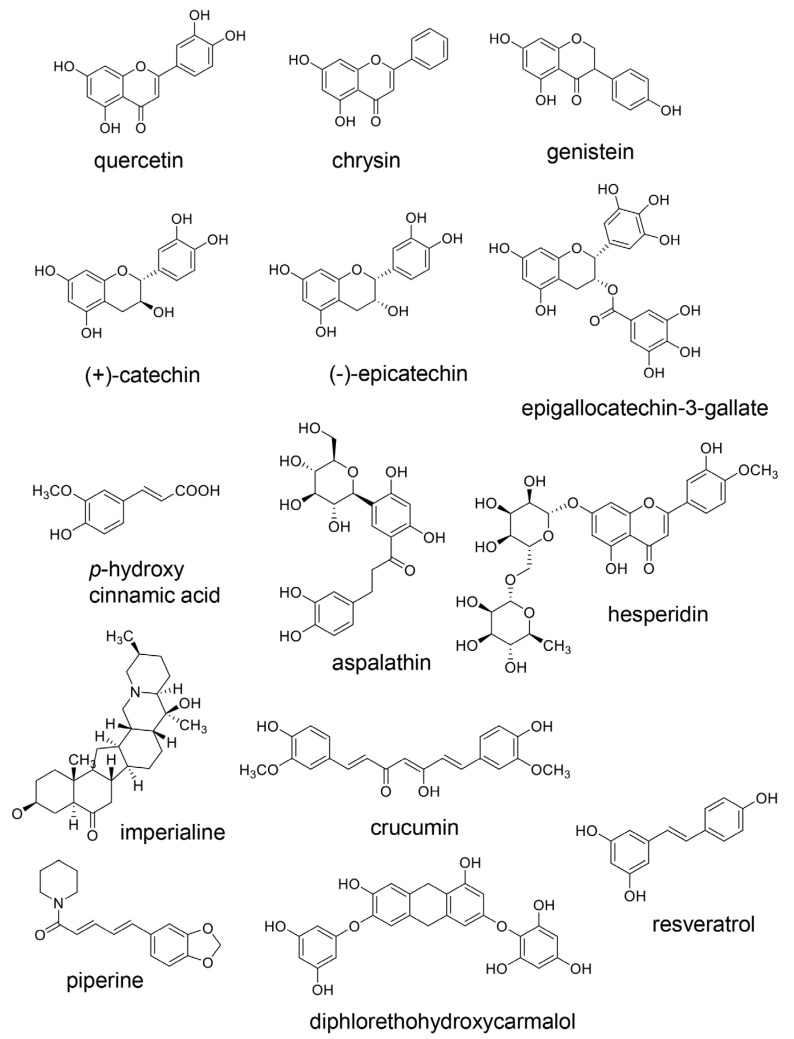
Natural compounds that have demonstrated the carbonyl trap effect and/or glyoxalase-1 (GLO)-1 activation. Quercetin [147], chrysin [148], genistein [149], (+)-catechin [152,153,154], (-)-epicatechin [152,153,154], epigallocatechin-3-gallate [150], *p*-hydroxy cinnamic acid [151], aspalathin [155], and hesperidin [156], imperialine [157], curcumin [158], piperine [158], diphlorethohydroxycarmalol [159], and resveratrol [158,160,161,162].

**Table 1 ijms-25-07319-t001:** Example of AGEs, their origin, and category.

Category	Origin	AGEs	Reference
AGE-1	Glucose	Pentosidine	[50]
AGE-2	Glyceraldehyde	Trihydroxy-triosidine	[48,49]
		GLAP	[48,49]
AGE-3	Glycolaldehyde	CML	[50]
		Hydroxymethyl-OP	[26]
AGE-4	Methylglyoxal	MG-H1	[48,49,50]
		Argpyrimidine	[48,49,50]
		CEL	[27]
AGE-5	Glyoxal	CML	[50]
		G-H1	[50]
AGE-6	3-deoxyglucosone	Pyralline	[50]
		Pentosidine	[50]

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
