# Peer review of "Generation and Accumulation of Various Advanced Glycation End-Products in Cardiomyocytes May Induce Cardiovascular Disease"

_ijms, 2024, doi:10.3390/ijms25137319_

Round 1

Reviewer 1 Report

Comments and Suggestions for Authors

The manuscript “Natural Compounds in Crude Drugs May Inhibit the Generation of Intracellular Advanced Glycation End-Products in Cardiomyocytes to Prevent Cardiovascular Disease” by Takata et al. reports various types of AGEs can be generated from saccharides (glucose and fructose) and their intermediate/non-enzymatic reaction byproducts. Although some preliminary results are demonstrated, the flow of this manuscript should be improved. Therefore, I would suggest authors may take at least a major revision. Here are the comments and suggestions:

1.      The title and abstract of this work should be revised.

2.      Some Tables should be added to compare the difference and reactions of AGEs.

3.      Some Figures can be combined.

4.      Authors are also suggested to add some schemes for easier understand the structures of AGEs.

5.      The conclusions should be extended, and what would be the perspectives of this work?

Comments on the Quality of English Language

1.      The title and abstract of this work should be revised.

Author Response

Response Letter to Reviewers’ Comments

Responses to Reviewer 1

Dear Reviewer 1:

Thank you for giving us the opportunity to submit a revised draft of our manuscript titled “Generation and Accumulation of Various Advanced Glycation End-Products in Cardiomyocytes May Induce Cardiovascular Disease” to the International Journal of Molecular Sciences (manuscript ID: 3067253). We appreciate the time and effort the reviewers have taken to provide their valuable feedback on our manuscript; their comments have enriched the manuscript and produced a more balanced account of our research. The manuscript has been revised by a professional English editor (Editage) to address all grammatical and syntax errors and improve the overall readability of the document.

We have inserted a New Table 1 and Figures 1, 2, 4–7, 13, 14. The Previous Figures 1, 2, 4–7, 13–16 in the first manuscript were removed.

We removed the reference 1, 42, 49, 96 which was substituted with a new one.

We have inserted the new references 163–165.

We corrected the Duplication of references (42 and 132).

The previous reference 49 was transported into 166.

Comments and Suggestions for Authors

The manuscript “Natural Compounds in Crude Drugs May Inhibit the Generation of Intracellular Advanced Glycation End-Products in Cardiomyocytes to Prevent Cardiovascular Disease” by Takata et al. reports various types of AGEs can be generated from saccharides (glucose and fructose) and their intermediate/non-enzymatic reaction byproducts. Although some preliminary results are demonstrated, the flow of this manuscript should be improved. Therefore, I would suggest authors may take at least a major revision. Here are the comments and suggestions:

Comment 1: The title and abstract of this work should be revised.

Response 1: We have rewritten the Title and Abstract. The revised title is “Generation and Accumulation of Various Advanced Glycation End-Products in Cardiomyocytes May Induce Cardiovascular Disease”.

Comment 2: Some Tables should be added to compare the difference and reactions of AGEs.

Response 2: We have inserted a New Table 1 which summarizes the classical categorization of AGEs, and describes the origin of AGEs (glucose, glyceraldehyde, glycolaldehyde, methylglyoxal, glyoxal, and 3-deoxyglucosone) and representative AGEs. The new table has been inserted in Section 4.1.

Comment 3: Some Figures can be combined.

Response 3: The previous Figures 5 and 6 have been combined (New Figure 6). The previous Figures 14, 15, and 16 have been combined (New Figure 14).

Comment 4: Authors are also suggested to add some schemes for easier understand the structures of AGEs.

Response 4: We have inserted New Figure 7 which shows glyceraldehyde generated MG-H1-, argpyrimidine, and GLAP-modified proteins in cells [71]. In Figure 7, each structure of AGEs was drawn and the structure of AGEs with lysine or arginine residue modifications of the proteins is shown.

Comment 5: The conclusions should be extended, and what would be the perspectives of this work?

Response 5: The future perspectives of this work include (i) the identification of low molecular compounds in traditional medicines that are able to inhibit the generation of intracellular AGEs in cardiomyocytes and prevent CVD and (ii) the identification of the underlying mechanisms of these compounds in their prevention of CVD.

Comments on the Quality of English Language

Comment 6: The title and abstract of this work should be revised.

Response 6: We have rewritten the title and abstract. The revised title is “Generation and Accumulation of Various Advanced Glycation End-Products in Cardiomyocytes May Induce Cardiovascular Disease”.

Reviewer 2 Report

Comments and Suggestions for Authors

The reviewed manuscript presents interestingly the involvement of AGEs in the pathogenesis of myocardial diseases. The text was prepared accessibly and the processes relevant to the issue at hand were illustrated schematically by the authors in numerous figures (16!), which facilitates understanding their intricacies without a doubt. However, the sections on plant inhibitors of glycation and cross-linking of biomolecules (mainly proteins and peptides) and compounds that trap reactive carbonyls unfortunately fall short of the others. Similarly, the manuscript title 'Natural Compounds in Crude Drugs May Inhibit the Generation of Intracellular Advanced Glycation End-Products in Cardiomyocytes to Prevent Cardiovascular Disease' seems mismatched with the main body of the manuscript.

1/ The full scientific binominal name of the plant species includes the citation, i.e. the author's name, e.g. Gynostemma pentaphyllum (Thunb.) Makino; please complete the names of all cited species.

2/ I suggest replacing the term 'crude drugs' with the more commonly used 'natural medicines/drugs' or 'traditional medicines/drugs'. Plant raw materials used in the manufacture of herbal (traditional) medicines are referred to in European regulations as 'plant substances', e.g., peppermint leaf (Menthae piperitae folium in Latin). Compounds separated from plant raw materials are called phytochemicals/phytoconstituents.

3/ Traditional herbal medicines are used extensively in the world, including in recommended therapies in Europe (European Pharmacopoeia, monographs of the European Medicines Agency, see https://www.ema.europa.eu/en/committees/committee-herbal-medicinal-products-hmpc) and the USA (botanicals=botanical drugs, FDA). Please include this information in section 7.1. 'The Traditional Medicines and Crude Drugs'.

4/ The herbal compounds listed in Section 7.3. 'Inhibition of the Generation of Intracellular AGEs' belongs to several chemical groups (listed rather randomly). They include flavonoids (quercetin, chrysin, genistein, aspalathin, hesperidin) known for their ability to trap reactive carbonyls but also other phenols/polyphenols (curcumin, resveratrol, epigallocatechin-3-gallate, etc.) or the alkaloid piperine. Although not mentioned by the authors, curcumin also can trap MGO (Hu, T. Y., Liu, C. L., Chyau, C. C., & Hu, M. L. (2012). Trapping of methylglyoxal by curcumin in cell-free systems and human umbilical vein endothelial cells. Journal of Agricultural and Food Chemistry, 60(33), 8190-8196). Of course, these are only examples; there are many more such compounds. On the other hand, numerous meta-analyses of epidemiological studies show that higher flavonoid intake is associated with reduced cardiovascular risk (e.g. Wang, X., Ouyang, Y. Y., Liu, J., & Zhao, G. (2014). Flavonoid intake and risk of CVD: a systematic review and meta-analysis of prospective cohort studies. British Journal of Nutrition, 111(1), 1-11; Li, T., Zhao, Y., Yuan, L., Zhang, D., Feng, Y., Hu, H., .... & Liu, J. (2024). Total dietary flavonoid intake and risk of cardiometabolic diseases: A dose-response meta-analysis of prospective cohort studies. Critical Reviews in Food Science and Nutrition, 64(9), 2760-2772).

5/ Lines 467 and 515 list malondialdehyde (MDA) and lysine-derived AGE among the glycation products identified in cardiomyocyte lysate. Peroxidation of polyunsaturated fatty acids produces MDA. Similarly, glyoxal or glyceraldehyde (from glycerol oxidation) may also be by-products of lipid metabolism. Could the authors also include AGEs from lipid metabolism in their considerations (section 4. AGEs)?

6/ I suggest correcting the content of the last sentence in the conclusions as it is incomprehensible: 'Natural compounds present in crude drugs can generate??? intracellular AGEs that can prevent cardiomyocyte dysfunction.

7/ Please check and correct the literature - spelling errors, e.g.

‘Litwinowicz, K.; Waszczuk, E.; Kuzan, A.; Branowicka-SzydeÅ‚KO, A.; Gostomska-Pampuch, K.; Naqporowski, P.; Gamian, A. Alcoholic Liver Disease Is Associated with Elevated Plasma Levels of Novel Advanced Glycation End-Products: A Preliminary Study. Nutrients 2022, 14, 5266.’ (lines 837-839)

It should be:

Litwinowicz, K., Waszczuk, E., Kuzan, A., Bronowicka-Szydełko, A., Gostomska-Pampuch, K., Naporowski, P., & Gamian, A. (2022). Alcoholic Liver Disease Is Associated with Elevated Plasma Levels of Novel Advanced Glycation End-Products: A Preliminary Study. Nutrients, 14(24), 5266.

Author Response

Response Letter to Reviewers’ Comments

Responses to Reviewer 2

Dear Reviewer 2:

Thank you for giving us the opportunity to submit a revised draft of our manuscript titled “Generation and Accumulation of Various Advanced Glycation End-Products in Cardiomyocytes May Induce Cardiovascular Disease” to International Journal of Molecular Sciences (manuscript ID: 3067253). We appreciate the time and effort the reviewers have taken to provide their valuable feedback on our manuscript; their comments have enriched the manuscript and produced a more balanced account of our research. The manuscript has been revised by a professional English editor (Editage) to address all grammatical and syntax errors and improve the overall readability of the document.

We have inserted a New Table 1 and Figures 1, 2, 4–7, 13, 14. The Previous Figures 1, 2, 4–7, 13–16 in the first manuscript were removed.

We removed the reference 1, 42, 49, 96 which was substituted with a new one.

We have inserted the new references 163–165.

We corrected the Duplication of references (42 and 132).

The previous reference 49 was transported into 166.

Comments and Suggestions for Authors

The reviewed manuscript presents interestingly the involvement of AGEs in the pathogenesis of myocardial diseases. The text was prepared accessibly and the processes relevant to the issue at hand were illustrated schematically by the authors in numerous figures (16!), which facilitates understanding their intricacies without a doubt. However, the sections on plant inhibitors of glycation and cross-linking of biomolecules (mainly proteins and peptides) and compounds that trap reactive carbonyls unfortunately fall short of the others. Similarly, the manuscript title 'Natural Compounds in Crude Drugs May Inhibit the Generation of Intracellular Advanced Glycation End-Products in Cardiomyocytes to Prevent Cardiovascular Disease' seems mismatched with the main body of the manuscript.

Comment 1:  The full scientific binominal name of the plant species includes the citation, i.e., the author's name, e.g. Gynostemma pentaphyllum (Thunb.) Makino; please complete the names of all cited species.

Response 1: We have provided the full scientific binominal name of the plant species “Amazuru” and “Amachazuru” in the Introduction.

Comment 2: I suggest replacing the term 'crude drugs' with the more commonly used 'natural medicines/drugs' or 'traditional medicines/drugs'. Plant raw materials used in the manufacture of herbal (traditional) medicines are referred to in European regulations as 'plant substances', e.g., peppermint leaf (Menthae piperitae folium in Latin). Compounds separated from plant raw materials are called phytochemicals/phytoconstituents.

Response 2: We have used the term “Traditional Medicine” throughout the manuscript. We have changed the title to “Generation and Accumulation of Various Advanced Glycation End-Products in Cardiomyocytes May Induce Cardiovascular Disease”.

Comment 3: Traditional herbal medicines are used extensively in the world, including in recommended therapies in Europe (European Pharmacopoeia, monographs of the European Medicines Agency, see https://www.ema.europa.eu/en/committees/committee-herbal-medicinal-products-hmpc) and the USA (botanicals=botanical drugs, FDA). Please include this information in section 7.1. 'The Traditional Medicines and Crude Drugs'.

Response 3: Accordingly, we have rewritten the Section 7.1. Traditional medicines are used extensively worldwide, including Europe, North and South America, Asia, Africa, and Australia. In addition, the plants, their place of origin, and the phytochemicals they contain have been discussed together with their potential to prevent or treat various diseases. This information was inserted in the Section 7.1.

Comment 4: The herbal compounds listed in Section 7.3. 'Inhibition of the Generation of Intracellular AGEs' belongs to several chemical groups (listed rather randomly). They include flavonoids (quercetin, chrysin, genistein, aspalathin, hesperidin) known for their ability to trap reactive carbonyls but also other phenols/polyphenols (curcumin, resveratrol, epigallocatechin-3-gallate, etc.) or the alkaloid piperine. Although not mentioned by the authors, curcumin also can trap MGO (Hu, T. Y., Liu, C. L., Chyau, C. C., & Hu, M. L. (2012). Trapping of methylglyoxal by curcumin in cell-free systems and human umbilical vein endothelial cells. Journal of Agricultural and Food Chemistry, 60(33), 8190-8196). Of course, these are only examples; there are many more such compounds. On the other hand, numerous meta-analyses of epidemiological studies show that higher flavonoid intake is associated with reduced cardiovascular risk (e.g. Wang, X., Ouyang, Y. Y., Liu, J., & Zhao, G. (2014). Flavonoid intake and risk of CVD: a systematic review and meta-analysis of prospective cohort studies. British Journal of Nutrition, 111(1), 1-11; Li, T., Zhao, Y., Yuan, L., Zhang, D., Feng, Y., Hu, H., .... & Liu, J. (2024). Total dietary flavonoid intake and risk of cardiometabolic diseases: A dose-response meta-analysis of prospective cohort studies. Critical Reviews in Food Science and Nutrition, 64(9), 2760-2772).

Response 4: Accordingly, we have added the details regarding the inhibitory activity of curcumin on the generation of intracellular AGEs with (i) the carbonyl trap system and (ii) the activation of GLO-1 in Section 7.3. Furthermore, we have inserted a new citation 163 (J. Agric. Food Chem. 2012, 60: 8190-8196).

Quercetin [147], chrysis [148], genistein [149], epigallocatechin-3-garate [150], (+)-catechin [152–154], (-)-epicatechin [152–154], aspalathin [155], hesperidin [156], and piperine [158] are flavonoids. In a meta-analysis of cohort studies, the intake of flavonoids prevented the risk of CVD. These intracellular AGEs may be involved in enhancing the dysfunction of cardiomyocytes within the context of different types of CVD and are beneficial for our investigation.

We have added this information to Section 7.3. and have inserted two new references 164 and 165 (Br. J. Nutr. 2014, 111: 1-111 and Crit. Rev. Food Sci. Nutr. 2024, 64: 2760-2772).

Comment 5: 467 and 515 list malondialdehyde (MDA) and lysine-derived AGE among the glycation products identified in cardiomyocyte lysate. Peroxidation of polyunsaturated fatty acids produces MDA. Similarly, glyoxal or glyceraldehyde (from glycerol oxidation) may also be by-products of lipid metabolism. Could the authors also include AGEs from lipid metabolism in their considerations (section 4. AGEs)?

Response 5: Methylglyoxal, glyoxal, glyceraldehyde, and glycolaldehyde are generated via lipid oxidation [Ref. 50]. They can also be generated from another route without the metabolism of saccharides. The information that AGEs are able to be derived via lipid oxidation is described in Section 4.1.

Since malondialdehyde (MDA) is derived via peroxidation of polyunsaturated fatty acids, MDA-modified AGEs can be generated/accumulated from lipid metabolism. We have included this information in Section 6.3.

Furthermore, we suggest that CEL, MG-H1, dihydroxy imidazoline, and p-hydroxyphenylglyoxal-arginine adducts derived from methylglyoxal and glyoxal, we describe their accumulation might be associated with lipid oxidation [Ref. 50,108–112] in Section 6.3.

Comment 6: I suggest correcting the content of the last sentence in the conclusions as it is incomprehensible: 'Natural compounds present in crude drugs can generate??? intracellular AGEs that can prevent cardiomyocyte dysfunction.

Response 6: We have corrected the sentences in the Conclusion.

Comment 7: Please check and correct the literature - spelling errors, e.g.

‘Litwinowicz, K.; Waszczuk, E.; Kuzan, A.; Branowicka-SzydeÅ‚KO, A.; Gostomska-Pampuch, K.; Naqporowski, P.; Gamian, A. Alcoholic Liver Disease Is Associated with Elevated Plasma Levels of Novel Advanced Glycation End-Products: A Preliminary Study. Nutrients 2022, 14, 5266.’ (lines 837-839)

It should be:

Litwinowicz, K., Waszczuk, E., Kuzan, A., Bronowicka-Szydełko, A., Gostomska-Pampuch, K., Naporowski, P., & Gamian, A. (2022). Alcoholic liver disease is associated with elevated plasma levels of novel advanced glycation end-products: a preliminary study. Nutrients, 14(24), 5266.

Response 7: We have corrected the citation.

Reviewer 3 Report

Comments and Suggestions for Authors

The review manuscript entitled "Natural Compounds in Crude Drugs May Inhibit the Generation of Intracellular Advanced Glycation End-Products in Cardiomyocytes to Prevent Cardiovascular Disease", is very interesting, and requires improvement in some aspects of its writing structure for which I present the following observations:

I suggest modifying the title of the review since the information content related to natural compounds in crude drugs is brief, and only corresponds to section 7. 

It is convenient to include in the initial part of the introduction a paragraph on information related to the incidence and prevalence of cardiovascular diseases worldwide.

Line 61: what type of protein is "action". I believe that the correct should be F-actin.

Figure 1 should be improved by adding in the scheme the names of the components that are part of the Figure.

There is no structural relationship between the myosin and F-actin-tropomyosin filaments (F-actin, tropomyosin, and troponin complex) in Figure 1 and Figure 2. The forms in which these structures are presented in each figure are very different from each other.

The legend of Figure 4 is repetitive with the names of the products of saccharides, which are shown in that figure.

Indicate in Figure 5, which structures correspond to MGO-AGEs and which structures correspond to GA-AGEs.

The legend of Figure 6 does not describe why MG-H1 and Argpyrimidine belong to MGO-AGEs and GA-AGEs.

It would be very interesting to include in section 4 (4. AGES) some example of Maillard reaction.

In section 4.4.3 Dietary AGEs, I suggest that they include a description of the absorption properties of dietary AGEs at the gastric and intestinal level.

In section 5. Various Types of Identification and Quantification Technologies for AGEs, the authors should describe the advantages and disadvantages of each technology for AGEs, especially, what type of AGEs (free-type AGEs or modified proteins) these technologies can quantify or identify. On the other hand, the description of the Fluorimetry technology (5.1) is too brief.

In line 162 the authors mention nuclear magnetic resonance (NMR) as a technology to identify AGEs, however, in section 5, the authors do not include its description.

In the legend of Figure 12 the authors should describe what kind of modification AGEs produce on RyR2 (does it increase or decrease its activity?).

Lines 564-566: Based on the following paragraph: "Papadaki et al. indicated that the glycation of myofilament proteins causes sarcomere dysfunction because the glycated myofilament reduces Ca2+ sensitivity and calcium-activated forces (Figure 13).", please indicate which is the physiological effect on the heart: does it affect the number of beats per minute or does it affect the force of the beats, or both?

In the title of the section "6.5 Generation of TAGE in Cardiomyocyte[11]", please delete the term "[11]".

In the conclusions section, I find the following paragraph confusing: "Natural compounds present in crude drugs can generate intracellular AGEs that can prevent cardiomyocyte dysfunction". However, according to section 7.3, the natural compounds present in crude drugs generate inhibition of intracellular AGEs.

Line 691: In the conclusions section the following is written: "Based on the results of this study, ...", to which study does it refer? The present work is a review.

Author Response

Response Letter to Reviewers’ Comments

Responses to Reviewer 3

Dear Reviewer 3:

Thank you for giving us the opportunity to submit a revised draft of our manuscript titled “Generation and Accumulation of Various Advanced Glycation End-Products in Cardiomyocytes May Induce Cardiovascular Disease” to the International Journal of Molecular Sciences (manuscript ID: 3067253). We appreciate the time and effort the reviewers have taken to provide their valuable feedback on our manuscript; their comments have enriched the manuscript and produced a more balanced account of our research. The manuscript has been revised by a professional English editor (Editage) to address all grammatical and syntax errors and improve the overall readability of the document.

We have inserted a New Table 1 and Figures 1, 2, 4–7, 13, 14. The Previous Figures 1, 2, 4–7, 13–16 in the first manuscript were removed.

We removed the reference 1, 42, 49, 96 which was substituted with a new one.

We have inserted the new references 163–165.

We corrected the Duplication of references (42 and 132).

The previous reference 49 was transported into 166.

Comments and Suggestions for Authors

The review manuscript entitled "Natural Compounds in Crude Drugs May Inhibit the Generation of Intracellular Advanced Glycation End-Products in Cardiomyocytes to Prevent Cardiovascular Disease", is very interesting, and requires improvement in some aspects of its writing structure for which I present the following observations:

Comment 1: I suggest modifying the title of the review since the information content related to natural compounds in crude drugs is brief, and only corresponds to section 7.

Response 1: The revised title is “Generation and Accumulation of Various Advanced Glycation End-Products in Cardiomyocytes May Induce Cardiovascular Disease”.

Comment 2: It is convenient to include in the initial part of the introduction a paragraph on information related to the incidence and prevalence of cardiovascular diseases worldwide.

Response 2: We didn’t the information related to the incidence and prevalence of CVD in the world. However, we inserted the New reference (Ref. 1) in Introduction (Section 1), and describe the information that the ration of CVD in the cause of death in China is approximately 40%.

Comment 3: Line 61: what type of protein is "action". I believe that the correct should be F-actin.

Response 3: We removed “action”, and the correct was “F-action”.

Comment 4: Figure 1 should be improved by adding in the scheme the names of the components that are part of the Figure.

Response 4:  Each component in cardiomyocytes was labeled.

Comment 5: There is no structural relationship between the myosin and F-actin-tropomyosin filaments (F-actin, tropomyosin, and troponin complex) in Figure 1 and Figure 2. The forms in which these structures are presented in each figure are very different from each other.

Response 5: The illustration in Figure 2, the fact that the myosin combines F-actin when calcium ion combine for troponin was visualized.

Comment 6: The legend of Figure 4 is repetitive with the names of the products of saccharides, which are shown in that figure.

Response 6: We removed the previous Figure 4, and inserted the New Figure 5. In this Figure 5, we introduced the information that “methylglyoxal, glyceraldehyde, and 3-deoxyglucosone are the origin of AGE-4, -2, and -6, respectively” .

Comment 7: Indicate in Figure 5, which structures correspond to MGO-AGEs and which structures correspond to GA-AGEs.

Response 7: We removed the previous Figures 5 and 6, and inserted the New Figure 6. In the New Figure 6, we introduced some MGO- and GA-AGEs which were categorized based on the classical categorization of them. We indicate that MG-H1 and Argpyrimidine belong both MGO- and GA-AGEs because they are able to generated from both methylglyoxal and glyceraldehyde.

Comment 8: The legend of Figure 6 does not describe why MG-H1 and Argpyrimidine belong to MGO-AGEs and GA-AGEs.

Response 8: We removed the previous Figures 5 and 6, and inserted the New Figure 6. In the New Figure 6, we introduced some MGO- and GA-AGEs which were categorized based on the classical categorization of them. We indicate that MG-H1 and Argpyrimidine belong both MGO- and GA-AGEs because they are able to generated from both methylglyoxal and glyceraldehyde.

Comment 9: It would be very interesting to include in section 4 (4. AGES) some example of Maillard reaction.

Response 9: We provided the New Figure 4 which introduce each CML is generated from some routes which contain Maillard reaction and other reactions (e.g. autooxidation) [Ref. 49,50]. This Figure is able to indicate each CML is generated from the various routes.

We inserted New reference 49 to explain these information.

We described these sentences in Section 4.1.

Furthermore, the routes of the generation of AGEs contained Maillard reaction which re-quire the generation of Schiff bases and Amadori products, and other reactions (e.g. au-tooxidation) [49,50]. We introduced the generation routes of CML (Figure 4). CML is able to be generated from Amadori products which origine is glucose (Figure 4). In contrast, glyoxal can be produced from autooxidation of glucose, and CML is able to be generated from it. Glycolaldehyde can be produced from Schiff bases, and generate CML. Because glycolaldehyde is origin of glyoxal, CML is able to be generated from some routes (Figure 4) [49,50].

Comment 10: In section 4.4.3 Dietary AGEs, I suggest that they include a description of the absorption properties of dietary AGEs at the gastric and intestinal level.

Response 10: We described the information of the oral intake, digestion, and absorption of dietary AGEs in the Section 4.4.3.

Furthermore, the effects of dietary AGEs for human body are affected the digestion and absorption because they are intake from mouth [50]. Though the digestion of dietary AGEs may start in oral epithelial cells by saliva enzyme such as α-amylase, most of them are directed to the gastric trac and intestinal phase where absorbance AGEs are delivered to the circulatory system and their remains are excreted in urine [50].

Comment 11: In section 5. Various Types of Identification and Quantification Technologies for AGEs, the authors should describe the advantages and disadvantages of each technology for AGEs, especially, what type of AGEs (free-type AGEs or modified proteins) these technologies can quantify or identify. On the other hand, the description of the Fluorimetry technology (5.1) is too brief.

Response 11: To explain the analysis of free-type of AGEs and AGEs-modified proteins with the Fluorimetry technology, we described some sentences in the Section 5.1.

“This characteristics of fluorimetry are that both free-type AGEs and AGEs-modified proteins are able to be quantified. AGEs were generally exited at wavelength of 370 nm and the fluorescence emitted at 440 nm. In the general characters of the analysis, each type of AGEs (e.g. MG-H1, argpyrimidine) are unable to be distinguished. However, Pinoto et a. reported that they quantified pentosidine because it is excited at a wavelength of 370 nm and fluorescence emitted at 378 nm [91].”

To explain the analysis of free type of AGEs and AGEs-modified protein with Immunostaining, western blotting, slot blotting, and ELISA, we inserted some sentences in Section 5.2.

“The respective advantages of these approaches are that anti-AGEs antibodies can recognize and target AGEs, but whose specific structures remain unclear. In contrast, their limitation of analysis is their inability to define the modification site on the AGE-modified protein (Figures 8 and 9). Furthermore, these technologies are suitable to analyze AG-Es-modified protein (or AGEs-modified peptide). Because the preparation of antibodies which can recognize and combine the low-molecular compounds is difficult, I believe that they are unsuitable the analysis of free-type of AGEs [48].”  

To explain the analysis of free type of AGEs and AGEs-modified protein with GC-MS, we inserted some sentences in Section 5.6.

“Therefore, the analysis of AGEs-modified proteins is unable to performed with GC-MS. Free type of AGEs are suitable to analysis with GC-MS. In contrast, researchers can be identified and quantified the free-type of AGEs which were prepared from acid hydrolysis of AGEs-modified protein [60,99,100]. However, free type of AGEs needs the esterification to prepare the sample in GC-MS because sample must be highly volatile [48].”

To explain the analysis of type of AGEs and AGEs-modified proteins with ESI-/MALDI-MS, we inserted some sentences in the Section 5.7.

“Under the step of acid hydrolysis of AGEs-modified protein, and analysis of free-type AGEs were able to be performed with ESI-/MALDI-MS as well as GC-MS [48]. Because the step of esterification of free-type AGEs don’t require to be performed in ESI-/MALDI-MS, this is beneficial of these technologies compared with GC-MS.”

“The limitation of ESI- and MALDI-MS is that automatic identification of both free-type AGEs and AGE-modified peptides is not possible if their structural data has not be in-putted into the database. Furthermore, we believe that type 2 multiple AGE patterns might be detected using automatic ESI- and MALDI-MS analysis because the analysis software recognizes a fragment ion peak in which AGE peptide structures might be modified by a single amino acid residue (Figure 9b). [71].”

Comment 12: In line 162 the authors mention nuclear magnetic resonance (NMR) as a technology to identify AGEs, however, in section 5, the authors do not include its description.

Response 12: We described the analysis of AGEs with NMR in the New Section 5.8.

“The novel free-type AGEs which were synthesized in tube and isolated from samples such as plasm were able to be identified with NMR. The data of signal of element (e.g. proton, carbon, nitrogen) indicate the combination of them (e.g. C-H, O-H, N-H) [55–59]. Once their structure was identified, the data of NMR can be used to identity ones which were isolated in samples. However, the detection of AGEs-modified proteins with NMR is difficult. Though signal of element proportional the existence of free-type AGEs, there is no report that quantification of AGEs in samples was performed with NMR.”

Comment 13: In the legend of Figure 12 the authors should describe what kind of modification AGEs produce on RyR2 (does it increase or decrease its activity?).

Response 13: The activity of RyR2 decreased, and the leak of calcium ion increased. We described this information in the legends of Figure 12.

Comment 14: Lines 564-566: Based on the following paragraph: "Papadaki et al. indicated that the glycation of myofilament proteins causes sarcomere dysfunction because the glycated myofilament reduces Ca2+ sensitivity and calcium-activated forces (Figure 13).", please indicate which is the physiological effect on the heart: does it affect the number of beats per minute or does it affect the force of the beats, or both?

Response 14:  Papadaki et al. indicated the possibility that the glycation of myofilament proteins causes sarcomere dysfunction because the glycated myofilament reduces Ca2+ sensitivity and cal-cium-activated forces in cardiomyocytes of patients with DM (Figure 13) [114]. However, the physiological effects (e.g. the number of beats per minute, force of the beats) in cardiac tissue didn’t be shown. Although they reported the reduction of Ca2+ sensitivity and calcium-activated forces in the skimmed cardiomyocytes which were treated with methyl-glyoxal in vitro, they didn’t identify and quantify the AGEs-modification [114]. The relationships between AGEs-modification in cardiomyocytes in the patients and the physio-logical effects should be researched in future. We described these sentences in the Section 6.4.

Furthermore, we revised the sentences in the legends of Figure 13.

“The predicted model that glycated myosin and F-actin-tropomyosin filament, and calcium-activated force [114].”

Comment 15: In the title of the section "6.5 Generation of TAGE in Cardiomyocyte[11]", please delete the term "[11]".

Response 15: We deleted [11] in Section 6.5.

Comment 16: In the conclusions section, I find the following paragraph confusing: "Natural compounds present in crude drugs can generate intracellular AGEs that can prevent cardiomyocyte dysfunction". However, according to section 7.3, the natural compounds present in crude drugs generate inhibition of intracellular AGEs.

Response 16: We corrected the Conclusion section.

Comment 17: Line 691: In the conclusions section the following is written: "Based on the results of this study, ...", to which study does it refer? The present work is a review.

Response 17: We corrected the Conclusion section.

Reviewer 4 Report

Comments and Suggestions for Authors

The authors discussed how intracellular advanced glycation end-products (AGEs) in cardiomyocytes can lead to cardiac tissue dysfunction, a non-ischemic type of cardiovascular disease (CVD). Various AGEs are formed from saccharides and their byproducts through non-enzymatic reactions. Identified AGEs like CML and MG-H1 in cardiomyocytes can cause excessive Ca2+ leakage and reduced contractile force, leading to dysfunction. It highlights how natural compounds found in traditional medicines, such as Kampo, can inhibit the generation of intracellular AGEs, potentially preventing CVDs. The paper emphasizes the importance of understanding the mechanisms by which natural compounds in crude drugs can prevent AGE-induced CVDs, suggesting a potential therapeutic approach for lifestyle-related diseases. Specific comments:

1.          The abstract mentions various natural compounds inhibiting AGE generation. Could you provide specific examples of these compounds?

2.          In section 1, the paper discusses various types of AGEs. It would be helpful to include a table summarizing these AGEs and their sources.

3.          The mechanisms by which natural compounds inhibit AGEs (carbonyl trap effect and glyoxalase 1 activation) are mentioned in the abstract. Can you elaborate on these mechanisms in the main text?

4.          Figures 1 and 2 illustrate the cardiomyocyte model and calcium-activated force. Ensure that the figures are clearly labeled and explained in the text.

5.          Section 5 discusses various identification and quantification technologies for AGEs. Please provide more details on the methodologies used and their respective advantages and limitations.

6.          In section 6.1, the study of intracellular CML in cardiomyocytes is mentioned. How do the results contribute to the understanding of cardiomyocyte dysfunction in LSRDs?

7.          Highlight the novel aspects of your research compared to existing studies on intracellular AGEs and cardiomyocyte dysfunction.

8.          Conclude with potential future research directions. Are there any other natural compounds or traditional medicines that could be explored for preventing CVDs related to intracellular AGEs?

Author Response

Response Letter to Reviewers’ Comments

Responses to Reviewer 4

Dear Reviewer 4:

Thank you for giving us the opportunity to submit a revised draft of our manuscript titled “Generation and Accumulation of Various Advanced Glycation End-Products in Cardiomyocytes May Induce Cardiovascular Disease” to International Journal of Molecular Sciences (manuscript ID: 3067253). We appreciate the time and effort the reviewers have taken to provide their valuable feedback on our manuscript; their comments have enriched the manuscript and produced a more balanced account of our research. The manuscript has been revised by a professional English editor (Editage) to address all grammatical and syntax errors and improve the overall readability of the document.

We have inserted a New Table 1 and Figures 1, 2, 4–7, 13, 14. The Previous Figures 1, 2, 4–7, 13–16 in the first manuscript were removed.

We removed the reference 1, 42, 49, 96 which was substituted with a new one.

We have inserted the new references 163–165.

We corrected the Duplication of references (42 and 132).

The previous reference 49 was transported into 166.

Comments and Suggestions for Authors

The authors discussed how intracellular advanced glycation end-products (AGEs) in cardiomyocytes can lead to cardiac tissue dysfunction, a non-ischemic type of cardiovascular disease (CVD). Various AGEs are formed from saccharides and their byproducts through non-enzymatic reactions. Previously identified AGEs like CML and MG-H1 in cardiomyocytes can cause excessive Ca2+ leakage and reduced contractile force, leading to dysfunction. It highlights how natural compounds found in traditional medicines, such as Kampo, can inhibit the generation of intracellular AGEs, potentially preventing CVDs. The paper emphasizes the importance of understanding the mechanisms by which natural compounds in crude drugs can prevent AGE-induced CVDs, suggesting a potential therapeutic approach for lifestyle-related diseases. Specific comments:

Comment 1: The abstract mentions various natural compounds inhibiting AGE generation. Could you provide specific examples of these compounds?

Response 1: Quercetin, curcumin and epigallocatechin-3-gallate have been included as examples of natural phytochemicals that inhibit AGEs in traditional medicines in the Abstract and Introduction section [Ref.46,47]

Comment 2: In section 1, the paper discusses various types of AGEs. It would be helpful to include a table summarizing these AGEs and their sources.

Response 2: We have inserted a New Table 1 which summarizes the classical categorization of AGEs, and describes the origin of AGEs (glucose, glyceraldehyde, glycolaldehyde, methylglyoxal, glyoxal, and 3-doxylucosone) and representative AGEs. The new table has been inserted in Section 4.1.

Comment 3: The mechanisms by which natural compounds inhibit AGEs (carbonyl trap effect and glyoxalase 1 activation) are mentioned in the abstract. Can you elaborate on these mechanisms in the main text?

Response 3: We have described both the carbonyl trap effect and glyoxal 1 activation in Sections 7.3. and 6.3.

Carbonyl trap: The precursors of AGEs have a carbonyl or aldehyde group, which can react with the amino acid residue (e.g. lysine, arginine) to generate AGEs. Compounds that can react with the carbonyl or aldehyde group of AGE precursors, inhibit the generation of AGEs [Ref. 143]. This is a very simple method to inhibit to generation of AGEs, however, such inhibitors (e.g., aminoguanidine) may induce cytotoxicity.

We have added this information to Section 7.3.

GLO-1 activation: Because GLO-1 can metabolize methylglyoxal and glyoxal, the activated GLO-1 contributes to their reduction and inhibits the generation of MGO-AGEs and GO-AGEs [Ref. 104,106,109].

We have added this information to Section 6.3.

Comment 4: Figures 1 and 2 illustrate the cardiomyocyte model and calcium-activated force. Ensure that the figures are clearly labeled and explained in the text.

Response 4: Figures 1 and 2 have been clearly labeled and explained in Section 1.

Comment 5: Section 5 discusses various identification and quantification technologies for AGEs. Please provide more details on the methodologies used and their respective advantages and limitations.

Response 5. To explain the analysis of free-type of AGEs and AGEs-modified proteins with the Fluorimetry technology, we described some sentences in the Section 5.1.

“This characteristics of fluorimetry are that both free-type AGEs and AGEs-modified proteins are able to be quantified. AGEs were generally exited at wavelength of 370 nm and the fluorescence emitted at 440 nm. In the general characters of the analysis, each type of AGEs (e.g. MG-H1, argpyrimidine) are unable to be distinguished. However, Pinoto et a. reported that they quantified pentosidine because it is excited at a wavelength of 370 nm and fluorescence emitted at 378 nm [91].”

To explain the analysis of free type of AGEs and AGEs-modified protein with Immunostaining, western blotting, slot blotting, and ELISA, we inserted some sentences in Section 5.2.

“The respective advantages of these approaches are that anti-AGEs antibodies can recognize and target AGEs, but whose specific structures remain unclear. In contrast, their limitation of analysis is their inability to define the modification site on the AGE-modified protein (Figures 8 and 9). Furthermore, these technologies are suitable to analyze AG-Es-modified protein (or AGEs-modified peptide). Because the preparation of antibodies which can recognize and combine the low-molecular compounds is difficult, I believe that they are unsuitable the analysis of free-type of AGEs [48].”  

To explain the analysis of free type of AGEs and AGEs-modified protein with GC-MS, we inserted some sentences in Section 5.6.

“Therefore, the analysis of AGEs-modified proteins is unable to performed with GC-MS. Free type of AGEs are suitable to analysis with GC-MS. In contrast, researchers can be identified and quantified the free-type of AGEs which were prepared from acid hydrolysis of AGEs-modified protein [60,99,100]. However, free type of AGEs needs the esterification to prepare the sample in GC-MS because sample must be highly volatile [48].”

To explain the analysis of type of AGEs and AGEs-modified proteins with ESI-/MALDI-MS, we inserted some sentences in the Section 5.7.

“Under the step of acid hydrolysis of AGEs-modified protein, and analysis of free-type AGEs were able to be performed with ESI-/MALDI-MS as well as GC-MS [48]. Because the step of esterification of free-type AGEs don’t require to be performed in ESI-/MALDI-MS, this is beneficial of these technologies compared with GC-MS.”

“The limitation of ESI- and MALDI-MS is that automatic identification of both free-type AGEs and AGE-modified peptides is not possible if their structural data has not be in-putted into the database. Furthermore, we believe that type 2 multiple AGE patterns might be detected using automatic ESI- and MALDI-MS analysis because the analysis software recognizes a fragment ion peak in which AGE peptide structures might be modified by a single amino acid residue (Figure 9b). [71].”

Comment 6. In section 6.1, the study of intracellular CML in cardiomyocytes is mentioned. How do the results contribute to the understanding of cardiomyocyte dysfunction in LSRDs?

Response 6: In this investigation, the intracellular role of CML in cardiomyocytes remains unclear. However, we can provide useful data to predict their functional activity. Mastrocola et al. reported that intracellular CML-modified proteins were generated/accumulated in skeletal muscle in C57B1/6J mice fed a high fructose diet, and they may induce lipogenesis [Ref. 76]. CML-modified proteins might induce lipogenesis in cardiomyocytes. We have described this information in Section 6.1.

Comment 7: Highlight the novel aspects of your research compared to existing studies on intracellular AGEs and cardiomyocyte dysfunction.

Response 7: In previous studies (e.g., immunostaining, western blotting, slot blotting, and ELISA), researchers revealed the existence of individual AGEs and the AGEs-modification of protein (e.g. RyR2) using anti-AGEs antibodies. These analyses are useful to identify crude AGE patterns (Figure 7). Furthermore, various types of AGE-modified proteins (Type 1 and 2 diverse AGE pattern, and type 1 multiple AGE pattern) can be identified (Figure 8,9b). AGE structures may induce dysfunction in combination, or a single AGE structure may prevent dysfunction. Based on the development of the technology that allows the analysis of AGE structures and their modification, further research will identify individual AGE-modified proteins in cardiomyocytes and will reveal the mechanisms of intracellular AGEs that induce CVD.

We have added a description in New Section 6.6.  

Comment 8: Conclude with potential future research directions. Are there any other natural compounds or traditional medicines that could be explored for preventing CVDs related to intracellular AGEs?

Response 8: The future perspectives of this work include (i) the identification of low molecular compounds in traditional medicines that are able to inhibit the generation of intracellular AGEs in cardiomyocytes and prevent CVD and (ii) the identification of the underlying mechanisms of these compounds in their prevention of CVD.

We have described these future perspectives in the Conclusion.

Round 2

Reviewer 1 Report

Comments and Suggestions for Authors

It seems more acceptable now, but some figures were duplicated.

Author Response

Response Letter to Reviewers’ Comments (Round 2)

Responses to Reviewer 1

Dear Reviewer 1:

Thank you for giving us the opportunity to submit a revised draft of our manuscript titled “Generation and Accumulation of Various Advanced Glycation End-Products in Cardiomyocytes May Induce Cardiovascular Disease” to the International Journal of Molecular Sciences (manuscript ID: 3067253). We appreciate the time and effort the reviewers have taken to provide their valuable feedback on our manuscript; their comments have enriched the manuscript and produced a more balanced account of our research. The manuscript has been revised by a professional English editor (Editage) to address all grammatical and syntax errors and improve the overall readability of the document.

Comment 1: It seems more acceptable now, but some figures were duplicated.

Response 1: Because the structures of 3-deoxyglucosone were designed in both Figure 5 and 10 in the previous manuscript (the First revised manuscript), we revised the Figure 10 to remove the structure of 3-deoxyglucosone.

More, the legends of Figure 10 were revised.

To explain the Compounds in order shown in the Figures 4–6, 10, we described “CML, 3-deoxtglucosone, argpyrimidine, pentosidine, pyraline, and GA-pyridine” in the Section 6.2.

Reviewer 2 Report

Comments and Suggestions for Authors

A few minor comments:

-Lines 75 and 76 - correct spelling of the names Parthenocissus tricuspidata (Siebold & Zucc.) Planch., and Gynostemma pentaphyllum (Thunb.) Makino 

- Lines 737 and 738 - AGE precursors contain aldehyde or ketone groups (both classified as carbonyls);

- Line 765 - piperine is an alkaloid (see Figure 14, it can be compared to quercetin structure) and cannot be classified as a flavonoid;

please correct.

Author Response

Response Letter to Reviewers’ Comments (Round 2)

Responses to Reviewer 2

Dear Reviewer 2:

Thank you for giving us the opportunity to submit a revised draft of our manuscript titled “Generation and Accumulation of Various Advanced Glycation End-Products in Cardiomyocytes May Induce Cardiovascular Disease” to the International Journal of Molecular Sciences (manuscript ID: 3067253). We appreciate the time and effort the reviewers have taken to provide their valuable feedback on our manuscript; their comments have enriched the manuscript and produced a more balanced account of our research. The manuscript has been revised by a professional English editor (Editage) to address all grammatical and syntax errors and improve the overall readability of the document.

We have inserted a New Figure 10 in this manuscript. The Previous Figure 10 in the First Revised manuscript was removed.

Comment 1: -Lines 75 and 76 - correct spelling of the names Parthenocissus tricuspidata (Siebold & Zucc.) Planch., and Gynostemma pentaphyllum (Thunb.) Makino

Response 1: We correct spelling of the names of them in the Introduction section (Section 1).

Comment 2: - Lines 737 and 738 - AGE precursors contain aldehyde or ketone groups (both classified as carbonyls);

Response 2: We correct the sentences in the Section 7.3.

“The precursors of AGEs contain aldehyde or ketone groups (both classified as carbonyls),”

“Compounds that can react with aldehyde or ketone groups of AGE precursors inhibit the generation of AGEs [143].”

We described these sentences in the Section 7.3.

Comment 3: - Line 765 - piperine is an alkaloid (see Figure 14, it can be compared to quercetin structure) and cannot be classified as a flavonoid;

Response 3: Based on the comments of Reviewer 2, we removed “piperine [158]” in the in the Line 765 of the Section 7.3. in the Previous manuscript.

We described the new sentence in the Section 7.3.

“Quercetin [147], chrysis [148], genistein [149], epigallocatechin-3-garate [150], (+)-catechin [152–154], (-)-epicatechin [152–154], aspalathin [155], and hesperidin [156] are flavonoids.”

Reviewer 4 Report

Comments and Suggestions for Authors

The authors discussed how intracellular advanced glycation end-products (AGEs) in cardiomyocytes can lead to cardiac tissue dysfunction, a non-ischemic type of cardiovascular disease (CVD). Various AGEs are formed from saccharides and their byproducts through non-enzymatic reactions. Identified AGEs like CML and MG-H1 in cardiomyocytes can cause excessive Ca2+ leakage and reduced contractile force, leading to dysfunction. It highlights how natural compounds found in traditional medicines, such as Kampo, can inhibit the generation of intracellular AGEs, potentially preventing CVDs. The paper emphasizes the importance of understanding the mechanisms by which natural compounds in crude drugs can prevent AGE-induced CVDs, suggesting a potential therapeutic approach for lifestyle-related diseases. The revision of the manuscript is improved, no additional comments.

Author Response

Response Letter to Reviewers’ Comments (Round 2)

Responses to Reviewer 4

Dear Reviewer 4:

Thank you for giving us the opportunity to submit a revised draft of our manuscript titled “Generation and Accumulation of Various Advanced Glycation End-Products in Cardiomyocytes May Induce Cardiovascular Disease” to the International Journal of Molecular Sciences (manuscript ID: 3067253). We appreciate the time and effort the reviewers have taken to provide their valuable feedback on our manuscript; their comments have enriched the manuscript and produced a more balanced account of our research. The manuscript has been revised by a professional English editor (Editage) to address all grammatical and syntax errors and improve the overall readability of the document.

Comment 1: The authors discussed how intracellular advanced glycation end-products (AGEs) in cardiomyocytes can lead to cardiac tissue dysfunction, a non-ischemic type of cardiovascular disease (CVD). Various AGEs are formed from saccharides and their byproducts through non-enzymatic reactions. Identified AGEs like CML and MG-H1 in cardiomyocytes can cause excessive Ca2+ leakage and reduced contractile force, leading to dysfunction. It highlights how natural compounds found in traditional medicines, such as Kampo, can inhibit the generation of intracellular AGEs, potentially preventing CVDs. The paper emphasizes the importance of understanding the mechanisms by which natural compounds in crude drugs can prevent AGE-induced CVDs, suggesting a potential therapeutic approach for lifestyle-related diseases. The revision of the manuscript is improved, no additional comments.

Response 1: Thank you for your evaluation against our review article. Since we rewrote our manuscript based on other Reviewers comments, we believe that the revised manuscript will be accepted for you (We consider that the quality of the revised manuscript will be better than the previous manuscript).
